# Deterministic Bounds and Random Estimates of Metric Tensors on Neuromanifolds

**Ke Sun**
CSIRO's Data61
Sydney, Australia
`Ke.Sun@data61.csiro.au`
`sunk@ieee.org`

## Abstract

The high-dimensional parameter space of deep neural networks — the neuromanifold — is endowed with a unique metric tensor defined by the Fisher information. Reliable and scalable computation of this metric tensor is valuable for theorists and practitioners. Focusing on neural classifiers, we return to a low-dimensional space of probability distributions, which we call the core space, and examine the spectrum and envelopes of its Fisher information matrix. We extend our discoveries there to deterministic bounds for the metric tensor on the neuromanifold. We introduce an unbiased random estimator based on Hutchinson's trace method and derive related bounds. It can be evaluated efficiently with a single backward pass per batch, with a standard deviation bounded by the true value up to scaling.

## 1 Introduction

Deep learning can be considered as a trajectory through *the space of neural networks* (*neuromanifold*; Amari 2016), where each point is a neural network instance with a prescribed architecture but different parameters. This work investigates classifier models in the form $p(y \mid x, \theta)$, where $x$ is the input features, $y \in \{1, \cdots, C\}$ is the class labels ($C \geq 2$), and $\theta \in \Theta$ is the network weights and biases. Given an unlabeled dataset $\mathcal{D}_x = \{x_1, x_2, \cdots\}$, the intrinsic structure of $\Theta$ is specified by the Fisher Information Matrix (FIM), defined as:

$$\mathcal{F}(\theta) := \sum_{x \in \mathcal{D}_x} \mathop{\mathbb{E}}_{p(y \mid x, \theta)} \left[ \frac{\partial \log p(y \mid x, \theta)}{\partial \theta} \frac{\partial \log p(y \mid x, \theta)}{\partial \theta^\top} \right] = \sum_{x \in \mathcal{D}_x} \mathop{\mathbb{E}}_{p(y \mid x, \theta)} \left[ \frac{\partial \ell_{xy}}{\partial \theta} \frac{\partial \ell_{xy}}{\partial \theta^\top} \right], \quad (1)$$

where $\ell_{xy}(\theta) := \log p(y \mid x, \theta)$ denotes the log-likelihood. This is based on a supervised model $x \to y$. For unsupervised models, one can treat $x$ as constant and apply the same formula. Under regularity conditions, $\mathcal{F}(\theta)$ is a $\dim(\theta) \times \dim(\theta)$ positive semidefinite (psd) matrix varying smoothly with $\theta \in \Theta$. Following Hotelling (1929), and independently Rao (1945), $\mathcal{F}(\theta)$ is used as a metric tensor on $\Theta$, representing a local, degenerate inner product[1]. For example, one can measure the intrinsic squared distance between $\theta$ and $\theta + \mathrm{d}\theta$, where $\mathrm{d}\theta$ is a small dynamic on $\Theta$, as $\mathrm{d}\theta^\top \mathcal{F}(\theta) \mathrm{d}\theta$.

The FIM is the unique metric tensor (Čencov, 1982) which underpins the *information geometry* of the neuromanifold $\Theta$ (Amari, 2016). The most widely used application of the FIM is perhaps geometry-inspired optimizers such as natural gradient (Amari, 1998), Adam (Kingma & Ba, 2015), and their variants (Martens & Grosse, 2015; Pascanu & Bengio, 2014; Yao et al., 2021; Lin et al., 2021). $\mathcal{F}$ is also applied to regularized fine-tuning (Lodha et al., 2023), pruning (Heskes, 2000; Tu et al., 2016), transfer learning (Chen et al., 2018), and overcoming catastrophic forgetting (Kirkpatrick et al., 2017). Theoretically, the FIM provides insights due to its connection with the Hessian of the loss landscape and generalization (Hochreiter & Schmidhuber, 1997), and that any $f$-divergence is locally characterized by the FIM (Blyth, 1994).

---

[*]Code: `https://github.com/sunk/fim-estimators`

[1]In the literature, $\mathcal{F}(\theta)$ is sometimes referred to as a curvature matrix (Martens, 2020). In rigorous terms, $\mathcal{F}(\theta)$ is a degenerate Riemannian metric tensor (psd with null directions), which belongs to the broad family of singular semi-Riemannian metrics (degenerate with an arbitrary signature; see Kupeli 1996; Sun & Nielsen 2025).

Given its deep and broad background, estimating $\mathcal{F}(\theta)$ with *theoretical guarantees on estimation quality* is important even in the absence of a specific application pipeline. Inaccurate estimates can lead to overly aggressive or overly conservative learning steps (Amari, 1998), or miscalculated saliency scores and suboptimal pruning decisions (Tu et al., 2016). In learning theory, a loosely estimated FIM undermines the validity of geodesic distances and the applicability of Cramér–Rao lower bounds, and may distort curvature-based sharpness, which is closely linked to generalization (Hochreiter & Schmidhuber, 1997). A widely used deterministic approximation is the empirical FIM (eFIM; a.k.a. empirical Fisher; see *e.g.* Le Roux et al. 2007) $\overline{\mathcal{F}}(\theta) \coloneqq \sum_{(x,y)\in\mathcal{D}} \left[\frac{\partial\ell_{xy}}{\partial\theta}\frac{\partial\ell_{xy}}{\partial\theta^\top}\right]$, where $\mathcal{D} = \{(x_1,y_1),(x_2,y_2),\cdots\}$ is a labeled dataset. Alternatively, the Monte Carlo (MC) estimator $\hat{\mathcal{F}}(\theta) = \frac{1}{m}\sum_{k=1}^m \frac{\partial\ell_{\hat{x}_k\hat{y}_k}}{\partial\theta}\frac{\partial\ell_{\hat{x}_k\hat{y}_k}}{\partial\theta^\top}$, where each $\hat{x}_k$ is drawn uniformly from $\mathcal{D}_x$ and $\hat{y}_k \sim p(y\,|\,\hat{x}_k)$, gives an unbiased stochastic estimate of $\mathcal{F}(\theta)$.

We advance the state-of-the-art in both deterministic and stochastic computations of the FIM, improving accuracy in terms of bound gap and variance. We make the following contributions: ① Envelopes of the FIM in the statistical simplex (space of output probabilities); ② Deterministic bounds of the FIM for classifier networks and their tightness analysis; ③ A novel family of random FIM estimators based on Hutchinson's trick (Hutchinson, 1990; Skorski, 2021), which can be computed efficiently with bounded variance; ④ An empirical study estimating the FIM of modern deep classifiers (*e.g.* DistilBERT; Sanh et al. 2019) to showcase the advantages of Hutchinson's estimate in real-world settings.

The remainder of this section introduces our notation. Section 2 develops fundamental bounds of the FIM in low dimensional spaces of probability distributions. Section 3 extends the deterministic bounds into the high dimensional neuromanifold. Section 4 introduces Hutchinson's FIM estimator and discusses its theoretical properties validated by numerical simulations. Section 5 positions our work into the literature. Section 6 concludes.

### NOTATIONS AND CONVENTIONS

We use lowercase letters such as $\lambda$ or $a$ for both vectors and scalars, which should be distinguished based on context, and capital letters such as $A$ for matrices. All vectors are column vectors. A scalar-vector or vector-scalar derivative such as $\partial\ell/\partial\theta$ yields a gradient vector of the same shape as the vector. A vector-vector derivative such as $\partial z/\partial\theta$ denotes the $\dim(z) \times \dim(\theta)$ Jacobian matrix of the mapping $\theta \to z$. $\|\cdot\|$ denotes the Euclidean norm for vectors and the Frobenius norm for matrices. $\|\cdot\|_\sigma$ denotes the spectral norm (maximum singular value) of matrices. The metric tensors (variants of FIM) are listed in table 1.

Table 1: Metric tensors. We use $\mathcal{I}$ / $\overline{\mathcal{I}}$ / $\hat{\mathcal{I}}$ / $\mathbb{I}$ for low-dimensional statistical manifolds and use $\mathcal{F}$ / $\overline{\mathcal{F}}$ / $\hat{\mathcal{F}}$ / $\mathbb{F}$ for neuromanifolds. We optionally use superscripts to indicate the associated parameter space. For example, $\mathcal{I}^\Delta$ and $\mathcal{F}^\Delta$ denote the metric tensors on the statistical simplex and on the space of neural networks with simplex-valued outputs, respectively.

| FIM | empirical FIM (eFIM) | Monte Carlo FIM (MC FIM) | Hutchinson FIM |
|---|---|---|---|
| $\mathcal{I}(z)$ / $\mathcal{F}(\theta)$ | $\overline{\mathcal{I}}(z)$ / $\overline{\mathcal{F}}(\theta)$ | $\hat{\mathcal{I}}(z)$ / $\hat{\mathcal{F}}(\theta)$ | $\mathbb{I}(z)$ / $\mathbb{F}(\theta)$ |

## 2 GEOMETRY OF LOW-DIMENSIONAL CORE SPACES

Consider a classifier network $p(y\,|\,x,\theta) \coloneqq p(y\,|\,z(x,\theta))$, where $z(x,\theta)$ is the last layer's linear output. Due to the chain rule, we plug $\frac{\partial\ell_{xy}}{\partial\theta} = \left(\frac{\partial z}{\partial\theta}\right)^\top \frac{\partial\ell_{xy}}{\partial z}$ into Eq. (1). Then, we can easily arrive at

$$\mathcal{F}(\theta) = \sum_{x\in\mathcal{D}_x} \left(\frac{\partial z}{\partial\theta}\right)^\top \cdot \mathcal{I}(z(x,\theta)) \cdot \frac{\partial z}{\partial\theta}, \qquad (2)$$

which is in the form of a Gauss-Newton matrix (Martens et al., 2010), or a pullback metric tensor (Sun, 2020)[2] from a low-dimensional statistical manifold with metric $\mathcal{I}(z)$, to the much higher-dimensional neuromanifold with metric $\mathcal{F}(\theta)$. In this section, we rediscover the geometrical structure of the low-dimensional statistical manifold, which we refer to as the *core space*, or simply the *core*.

In multi-class classification, $y$ (given a feature vector $x$) follows a categorical distribution $p(y = i \,|\, x, \theta) = p_i(x, \theta)$, $i = 1, \cdots, C$. All possible categorical distributions over $\{1, \cdots, C\}$ form a closed statistical simplex $\Delta^{C-1} := \left\{ (p_1, \cdots, p_C) \ : \ \sum_{i=1}^{C} p_i = 1; \ \forall i, p_i \geq 0 \right\}$. The superscript $C - 1$ denotes the dimensionality of $\Delta$ and can be omitted. If $p \in \text{int}(\Delta^{C-1})$ (interior of $\Delta^{C-1}$), we can reparameterize $p = \text{SoftMax}(z)$, where $z \in \Re^C$ is the logits. The core $\Delta^{C-1}$ is a curved space, where $p$ or $z$ serves as a coordinate system, in the sense that different choices of $p$ or $z$ yield different distributions. By Eq. (1), the FIM is:

$$\mathcal{I}^{\Delta}(z) = \mathbb{E}\left[(e_y - p)(e_y - p)^{\top}\right] = \text{diag}\,(p) - pp^{\top}, \tag{3}$$

where $\text{diag}\,(\cdot)$ means the diagonal matrix constructed with a given diagonal vector. Below, depending on context, $\text{diag}\,(\cdot)$ also denotes a diagonal vector extracted from a square matrix. $e$ (without subscripts) denotes a vector of all ones, $e_y$ denotes the one-hot vector with only the $y$'th bit activated, and $e_{ij}$ denotes the binary matrix with only the $ij$'th entry set to 1. Note that $z$ is a redundant coordinate system as $\dim(z) = C > C - 1$. If $z \in \text{int}(\Delta^{C-1})$, $\mathcal{I}^{\Delta}(z)$ has a one-dimensional kernel: one can easily verify $\mathcal{I}^{\Delta}(z)(te) = 0$ for all $t \in \Re$.

Noting that $\mathcal{I}^{\Delta}(z)$ is a rank-1 perturbation of the diagonal matrix $\text{diag}\,(p)$, we can apply Cauchy's interlacing theorem and study the spectral properties of $\mathcal{I}^{\Delta}(z)$.

**Theorem 1** (Spectrum of Simplex FIM). *Assume the spectral decomposition $\mathcal{I}^{\Delta}(z) = \sum_{i=1}^{C} \lambda_i v_i v_i^{\top}$, where $\lambda_1 \leq \cdots \leq \lambda_C$. Then $\lambda_1 = 0$; $v_1 = e/\|e\|$; $\sum_{i=1}^{C} \lambda_i = 1 - \|p\|^2$; and*

$$\max\left\{p_i(1 - p_i)\right\} \cup \left\{p_{(C-1)}, \frac{1 - \|p\|^2}{C - 1}\right\} \leq \lambda_C \leq \min\left\{p_{(C)}, \ 2\max_i(p_i(1 - p_i)), \ 1 - \|p\|^2\right\},$$

*where $p_{(C-1)}$ and $p_{(C)}$ denote the second-largest and the largest elements of $p$, respectively.*

The largest eigenvalue of $\mathcal{I}^{\Delta}(z)$, denoted as $\lambda_C$, and its associated eigenvector correspond to the "most informative" direction at any $z \in \Delta^{C-1}$. By Theorem 1, $\lambda_C$ can be bounded from above and below. The bound gap is at most $\min\{p_{(C)} - p_{(C-1)}, \max_i(p_i(1 - p_i))\}$. We have found through numerical simulations that, in practice, the bounds in Theorem 1 are tight and can provide an estimate of $\lambda_C$ within a narrow range. The lemma below gives lower and upper bounds of $\mathcal{I}^{\Delta}(z)$, both with a simpler structure than $\mathcal{I}^{\Delta}(z)$, in the space of psd matrices under the Löwner (Loewner) partial order.

**Lemma 2.** $\forall z \in \text{int}(\Delta^{C-1})$, *assume the spectral decomposition $\mathcal{I}^{\Delta}(z) = \sum_{i=1}^{C} \lambda_i v_i v_i^{\top}$, where $\lambda_1 \leq \cdots \leq \lambda_{C-1} < \lambda_C$. Then, $\lambda_C v_C v_C^{\top} \preceq \mathcal{I}^{\Delta}(z) \preceq \text{diag}\,(p)$. Moreover, $\lambda_C v_C v_C^{\top}$ is the best rank-1 representation of $\mathcal{I}^{\Delta}(z)$ in the sense that no rank-1 matrix $B \neq \lambda_C v_C v_C^{\top}$ satisfies $\lambda_C v_C v_C^{\top} \preceq B \preceq \mathcal{I}^{\Delta}(z)$. Meanwhile, $\text{diag}\,(p)$ is the best diagonal representation of $\mathcal{I}^{\Delta}(z)$ in the sense that no diagonal matrix $D \neq \text{diag}\,(p)$ satisfies $\mathcal{I}^{\Delta}(z) \preceq D \preceq \text{diag}\,(p)$.*

The simplex FIM is upper-bounded by a diagonal matrix and lower-bounded by a rank-1 matrix. By Lemma 2, $\lambda_C v_C v_C^{\top}$ is the *lower envelope* (greatest lower bound) of $\mathcal{I}^{\Delta}(z)$ in rank-1 matrices, and $\text{diag}\,(p)$ is the *upper envelope* (least upper bound) of $\mathcal{I}^{\Delta}(z)$ in diagonal matrices. If the bounds in Lemma 2 are used as a deterministic estimate of $\mathcal{I}^{\Delta}(z)$, the error can be controlled, as shown below.

**Lemma 3.** *We have $\forall z \in \Delta$, $\|\mathcal{I}^{\Delta}(z) - \text{diag}\,(p)\| = \|p\|^2 \geq \frac{1}{C}$; meanwhile, $\|\mathcal{I}^{\Delta}(z) - \lambda_C v_C v_C^{\top}\| \leq \min\left\{1 - \|p\| - p_{(C-1)}, \sqrt{\sum_{i=2}^{C-1} p_{(i)}^2}\right\}$, where $p_{(i)}$ denotes the entries of $p$ sorted in ascending order.*

Note $\sqrt{\sum_{i=2}^{C-1} p_{(i)}^2}$ is the Euclidean norm of *trimmed* $p$, *i.e.* the vector obtained by removing $p$'s smallest and largest elements. By Lemma 3, the upper bound $\text{diag}\,(p)$ always incurs an error of at

---

[2]Strictly speaking, the pullback tensor requires the Jacobian of $\theta \to z$ to have full column rank everywhere, which is not satisfied in typical settings of deep neural networks. This leads to degenerate (singular) metric tensors.

least $1/C$. Depending on $p$, the lower bound $\lambda_C v_C v_C^\top$ can more accurately estimate $\mathcal{I}^\Delta(z)$ as the error can go to zero.

Alternatively, one can use random matrices to estimate $\mathcal{I}^\Delta(z)$. By Eq. (3), the rank-1 matrix $R(y) = (e_y - p)(e_y - p)^\top$ is an unbiased estimator of $\mathcal{I}^\Delta(z)$. The MC FIM of $\Delta$ is $\hat{\mathcal{I}}^\Delta(z) = \frac{1}{m} \sum_{k=1}^{m} R(\hat{y}_k)$, where each $\hat{y}_k \sim p(y \,|\, z)$. The associated eFIM is $\overline{\mathcal{I}}^\Delta(z) = R(y)$, where $y$ is a given empirical sample. The lemma below shows the worst case error of $\overline{\mathcal{I}}^\Delta(z)$.

**Lemma 4.** $\forall z \in \Delta^{C-1}$, $\exists y \in \{1, \cdots, C\}$, such that $\|R(y) - \mathcal{I}^\Delta(z)\| \geq 1 + \|p\|^2 - \lambda_C - 2p_{(1)}$ $\geq 2\|p\|^2 - 2p_{(1)}$.

The first "$\geq$" is tighter but the second "$\geq$" is easier to interpret. The term $\|p\|$ can be as large as 1 (when $p$ is close to one-hot). In such cases, using $R(y)$ to estimate $\mathcal{I}^\Delta(z)$ may incur significant error if $y$ is adversarially chosen.

In classification tasks with multiple binary labels, we assume $p(y_i = 1 \,|\, x) = p_i$ ($i = 1, \cdots, C$) and that all dimensions of $y$ are conditionally independent given $x$. All such distributions form a $C$-dimensional hypercube $\mathcal{C}^C(p) = \{(p_1, \cdots, p_C) : \forall i, 0 \leq p_i \leq 1\}$, which is the product space of 1-dimensional simplices. Consider $p_i = \sigma(z_i) := 1/(1 + \exp(-z_i))$ for $i = 1, \cdots, C$. In this case, the FIM is a diagonal matrix, given by

$$\mathcal{I}^C(z) = \mathrm{diag}\left((p_1(1 - p_1), \cdots, p_C(1 - p_C))\right) = \mathrm{diag}\left(\sigma'(z_1), \cdots, \sigma'(z_C)\right). \qquad (4)$$

In what follows, unless stated otherwise, our results pertain to the core $\Delta$ as it is more commonly used and has a more complex FIM as compared to $\mathcal{C}$.

## 3 FIM FOR CLASSIFIER NETWORKS: DETERMINISTIC ANALYSIS

In this section, we give lower and upper bounds of $\mathcal{F}^\Delta(\theta)$ (Proposition 5) and analyze each bound gap (Propositions 7 and 8). Our bounds result from simple matrix analysis and are more operational than related theoretical bounds such as monotonicity of the FIM under marginalization or coarse-graining (Amari, 2016). Our bounds are novel in that ① they are built on envelopes (tightest bound) in the core, and ② they depend on the order statistics of the output probability vector.

### 3.1 DETERMINISTIC LOWER AND UPPER BOUNDS

By Eq. (2), the neuromanifold FIM $\mathcal{F}(\theta)$ is determined by both the core space and the parameter-output Jacobian $\frac{\partial z}{\partial \theta}$. Similar to Lemma 2, we can have lower and upper bounds of $\mathcal{F}^\Delta(\theta)$ in the space of psd matrices (although these bounds are not envelopes as in Lemma 2).

**Proposition 5.** *If $p(y \,|\, x, \theta) \in \Delta^{C-1}$ is categorical, then $\forall \theta \in \Theta$, we have*

$$\sum_{x \in \mathcal{D}_x} \sum_{i=C-k+1}^{C} \lambda_i \left(\frac{\partial z}{\partial \theta}\right)^\top v_i v_i^\top \frac{\partial z}{\partial \theta} \preceq \mathcal{F}^\Delta(\theta) \preceq \sum_{x \in \mathcal{D}_x} \sum_{y=1}^{C} p(y \,|\, x, \theta) \frac{\partial z_y}{\partial \theta} \left(\frac{\partial z_y}{\partial \theta}\right)^\top$$

*for all $k \in \{1, \cdots, C-1\}$, where $\lambda_i := \lambda_i(x, \theta)$ and $v_i := v_i(x, \theta)$ denote the $i$-th eigenvalue and eigenvector of $\mathcal{I}(z(x, \theta))$, ordered such that $\lambda_1 \leq \lambda_2 \leq \cdots \leq \lambda_C$.*

**Remark.** *The LHS is a sum of $|\mathcal{D}_x|$ (number of samples in $\mathcal{D}_x$) matrices, each of rank $k$. Its rank is at most $k|\mathcal{D}_x|$. The RHS is a sum of $C|\mathcal{D}_x|$ matrices of rank-1 and potentially has a larger rank.*

**Remark.** *By Theorem 1, $\lambda_1 = 0$. Therefore, the first "$\preceq$" turns into "$=$" when $k = C-1$.*

If $p(y \,|\, x)$ is in $\mathcal{C}$, then $\mathcal{I}^C(z(x, \theta))$ is diagonal as in Eq. (4). By Eq. (2), we have $\mathcal{F}^C(\theta) = \sum_{x \in \mathcal{D}_x} \sum_{y=1}^{C} p_y(1 - p_y) \frac{\partial z_y}{\partial \theta} \left(\frac{\partial z_y}{\partial \theta}\right)^\top$, which is similar to the upper bound in Proposition 5. In summary, $\mathcal{F}(\theta)$ can be bounded or computed using the Jacobian $\frac{\partial z}{\partial \theta}$ as well as the output probabilities $p(y \,|\, x, \theta)$. The following analysis depends on the spectral properties of $\frac{\partial z}{\partial \theta}$. Across our statements, we denote the singular values of $\frac{\partial z}{\partial \theta}$, sorted in ascending order, as $\sigma_1(x, \theta) \leq \cdots \leq \sigma_C(x, \theta)$. In Proposition 5, by taking the trace on all sides, the trace of the FIM can be bounded from above and below.

**Corollary 6.** *If $p(y \mid x, \theta) \in \Delta^{C-1}$ is categorical, then it holds for all $\theta \in \Theta$ that*

$$\sum_{x \in \mathcal{D}_x} \lambda_C(x, \theta) \sigma_1^2(x, \theta) \leq \sum_{x \in \mathcal{D}_x} \sum_{i=2}^{C} \lambda_i(x, \theta) \sigma_{C+1-i}^2(x, \theta) \leq \operatorname{tr}(\mathcal{F}^{\Delta}(\theta)) \leq \sum_{x \in \mathcal{D}_x} \sum_{y=1}^{C} p(y \mid x, \theta) \left\| \frac{\partial z_y}{\partial \theta} \right\|^2.$$

These bounds are useful to get the overall scale of $\mathcal{F}^{\Delta}(\theta)$ without computing its exact value. The proposition below gives the error of the upper bound in Proposition 5 in terms of the Frobenius norm.

**Proposition 7** (Tightness of the Upper Bound)**.** *We have $\forall \theta \in \Theta$ that*

$$\sqrt{\sum_{x \in \mathcal{D}_x} \left\| \left( \frac{\partial z}{\partial \theta} \right)^{\top} p(x, \theta) \right\|^4} \leq \left\| \sum_{x \in \mathcal{D}_x} \sum_{y=1}^{C} p(y \mid x, \theta) \left( \frac{\partial z_y}{\partial \theta} \right)^{\top} \frac{\partial z_y}{\partial \theta} - \mathcal{F}^{\Delta}(\theta) \right\| \leq \sum_{x \in \mathcal{D}_x} \|p(x, \theta)\|^2 \sigma_C^2(x, \theta),$$

*where $p(x, \theta) = \operatorname{SoftMax}(z(x, \theta))$ denotes the output probability vector.*

We use the Frobenius norm for matrices but it is not difficult to bound the spectral norm using similar techniques. By Proposition 7, the error of the upper bound scales with the operator 2-norm (maximum singular value) of the parameter-output Jacobian $\frac{\partial z}{\partial \theta}$. As in the core space, the FIM upper bound remains loose. For example, let $p$ tend to be one-hot, the LHS in Proposition 7 does not vanish, but scales with certain rows of $\frac{\partial z}{\partial \theta}$ corresponding to the predicted $y$. Naturally, we also want to examine the error of the lower bound in Proposition 5, as detailed below.

**Proposition 8** (Tightness of the Lower Bound)**.** *We have for all $\theta \in \Theta$ and all $k \in \{1, \cdots, C-1\}$ that*

$$\left\| \sum_{x \in \mathcal{D}_x} \sum_{i=C-k+1}^{C} \lambda_i \left( \frac{\partial z}{\partial \theta} \right)^{\top} v_i v_i^{\top} \frac{\partial z}{\partial \theta} - \mathcal{F}^{\Delta}(\theta) \right\| \leq \sum_{x \in \mathcal{D}_x} \sqrt{\sum_{i=2}^{C-k} \sigma_{i+k}^4(x, \theta) p_{(i)}^2(x, \theta)}.$$

Clearly, as $p$ approaches a one-hot vector, all elements in the trimmed vector $p_{(i)}$, for $i = 2, \cdots, C-1$, tend to zero, and the error approaches zero since its upper bound on the RHS goes to zero. From this view, the lower bound in Proposition 5 is a better estimate as compared to the upper bound.

**Remark.** *By noting that $0 \leq \sigma_i(x, \theta) \leq \sigma_C(x, \theta)$, we relax the bound in Proposition 8 and get*

$$\left\| \sum_{x \in \mathcal{D}_x} \sum_{i=C-k+1}^{C} \lambda_i \left( \frac{\partial z}{\partial \theta} \right)^{\top} v_i v_i^{\top} \frac{\partial z}{\partial \theta} - \mathcal{F}^{\Delta}(\theta) \right\| \leq \sum_{x \in \mathcal{D}_x} \sqrt{\sum_{i=2}^{C-k} p_{(i)}^2(x, \theta) \cdot \sigma_C^2(x, \theta)}.$$

*The estimation error of the low-rank lower bound in Proposition 5 is controlled by the norms of the Jacobian and the trimmed probabilities $(p_{(2)}, \cdots, p_{(C-k)})$. The latter is upper bounded by $p_{(C-k)}(x, \theta)$, the $(k+1)$-th largest probability of each sample $x$. By comparing with the second "$\leq$" in Proposition 7, one can easily observe that Proposition 8 is tighter in general.*

## 3.2 EMPIRICAL FIM (EFIM)

Recall from the introduction that the eFIM $\overline{\mathcal{F}}(\theta)$ gives a biased, deterministic estimate of $\mathcal{F}(\theta)$. Intuitively, when the network is trained, computations based on the given labels are close to the expectation w.r.t. $p(y \mid x, \theta)$, and $\overline{\mathcal{F}}(\theta)$ is expected to approximate $\mathcal{F}(\theta)$ well. However, the bias of $\overline{\mathcal{F}}(\theta)$ can be enlarged if $y$ is set adversarially. By simple derivations, $\overline{\mathcal{F}}(\theta) = \sum_{x \in \mathcal{D}_x} \left( \frac{\partial z}{\partial \theta} \right)^{\top} \cdot R(y) \cdot \frac{\partial z}{\partial \theta}$. Observe that it is similar to Eq. (2), except $\mathcal{I}(z(x, \theta))$ is replaced by its empirical counterpart $R(y)$. If the neural network output is in $\Delta$, the error of eFIM can be bounded, as stated below.

**Proposition 9.** *$\forall \theta \in \Theta$, $\forall y$, we have $\|\mathcal{F}^{\Delta}(\theta) - \overline{\mathcal{F}}^{\Delta}(\theta)\|_{\sigma} \leq \sum_{x \in \mathcal{D}_x} (1 + \|p(x, \theta)\|^2) \sigma_C^2(x, \theta)$.*

Here we need to switch to the spectral norm $\| \cdot \|_{\sigma}$ to get a simple expression of the upper bound. The approximation error of $\overline{\mathcal{F}}^{\Delta}(\theta)$ is controlled by the spectral norm of the parameter-output Jacobian. The error in the Frobenius norm is even larger. The bound is loose as compared to Propositions 7 and 8.

We have found in Lemma 4 that using $R(y)$ to approximate $\mathcal{I}^{\Delta}(z)$ suffers from a large error if $y$ is chosen in a tricky way. The same principle applies to using $\overline{\mathcal{F}}(\theta)$ to approximate $\mathcal{F}(\theta)$.

**Proposition 10.** $\forall \theta \in \Theta$, $\forall x$, $\exists y$, such that

$$\left\| \left( \frac{\partial z}{\partial \theta} \right)^{\top} \mathcal{I}^{\Delta}(z(x,\theta)) \frac{\partial z}{\partial \theta} - \left( \frac{\partial z}{\partial \theta} \right)^{\top} R(y) \frac{\partial z}{\partial \theta} \right\|_{\sigma} \geq \sigma_1^2(x,\theta) \left| 1 + \|p(x,\theta)\|^2 - \lambda_C(x,\theta) - 2p_{(1)}(x,\theta) \right|.$$

In the above inequality, the LHS is the error of $\overline{\mathcal{F}}(\theta)$ for $x \in \mathcal{D}_x$. Therefore, when $y$ is set unfavorably, the eFIM suffers from an approximation error that scales with the smallest singular value of $\frac{\partial z}{\partial \theta}$. Among the investigated deterministic approximations of $\mathcal{F}^{\Delta}$, the lower bound in Proposition 5 provides the smallest guaranteed error but is relatively expensive to compute. We solve the computational issues in the next section.

## 4 HUTCHINSON'S ESTIMATE OF THE FIM

### 4.1 LIMITATIONS OF MONTE CARLO ESTIMATES

The quality of the MC estimate $\hat{\mathcal{F}}(\theta)$ can be arbitrarily bad. Consider the single-neuron model $z = \theta x$ for binary classification, where $z, \theta, x$ are all scalars, and $\theta$ is close to zero. Then $p \approx \frac{1}{2}$ is a fair Bernoulli distribution. $\mathcal{I}(z) = p(1-p) \approx \frac{1}{4}$. The Jacobian is simply $\frac{\partial z}{\partial \theta} = x$, and $\mathcal{F}(\theta) = \mathbb{E}_{p(x)} \left[ \frac{\partial z}{\partial \theta} \mathcal{I}(z) \frac{\partial z}{\partial \theta} \right] \approx \frac{1}{4} \mathbb{E}_{p(x)}[x^2]$. A basic MC estimator takes the form $\hat{\mathcal{F}}(\theta) = \frac{1}{4m} \sum_{i=1}^{m} x_i^2$, where $x_i$'s are independently and identically distributed according to $p(x)$. Its variance is $\mathrm{Var}(\hat{\mathcal{F}}) = \frac{1}{4m} [\mathbb{E}_{p(x)}(x^4) - \mathbb{E}_{p(x)}^2(x^2)]$. We let $p(x)$ be a heavy-tailed distribution, e.g. Student's $t$-distribution with $\nu > 4$ degrees of freedom, so that $\mathrm{Var}(\hat{\mathcal{F}})$ is large while $\mathcal{F}(\theta)$ is small. Then $\mathbb{E}_{p(x)}(x^2) = \frac{\nu}{\nu-2}$ and $\mathbb{E}_{p(x)}(x^4) = \frac{3\nu^2}{(\nu-2)(\nu-4)}$. The ratio $\frac{\mathbb{E}_{p(x)}(x^4)}{(\mathbb{E}_{p(x)} x^2)^2} = \frac{3(\nu-2)}{\nu-4}$ can be arbitrarily large when $\nu \to 4^+$. Therefore the coefficient of variation (CV) $\mathrm{Std}(\hat{\mathcal{F}})/\mathcal{F}(\theta)$ is unbounded. Throughout our analysis, the *CV is a key indicator* of the quality of a FIM estimator, as a bounded CV for a non-negative random variable $X$ ensures the random estimator's probability mass within $[0, \alpha\mu]$, where $\alpha > 1$ and $\mu \geq 0$ is the mean of $X$. If $\mathrm{CV} = \frac{\mathrm{Std}X}{\mu} \leq K$, then by Cantelli inequality, we have

$$\mathbb{P}(X \geq \alpha\mu) = \mathbb{P}(X \geq \mu + (\alpha-1)\mu) \leq \mathbb{P}\left( X \geq \mu + \frac{\alpha-1}{K}\mathrm{Std}X \right) \leq \left( 1 + \left( \frac{\alpha-1}{K} \right)^2 \right)^{-1}.$$

The general case is more complicated, but follows a similar idea. The variance of MC estimators depends on the 4th moment of the Jacobian $\frac{\partial z}{\partial \theta}$ w.r.t. $p(x)$ while the mean value $\mathcal{F}(\theta)$ only depends on the 2nd moment of $\frac{\partial z}{\partial \theta}$. The ratio of the MC estimator's variance to $\mathcal{F}^2(\theta)$, or the CV $\mathrm{Std}(\hat{\mathcal{F}})/\mathcal{F}(\theta)$, is unbounded without further assumptions on $p(x)$. One can increase the number of samples $m$ to reduce variance. However, this is computationally expensive especially in online settings.

### 4.2 HUTCHINSON'S ESTIMATE

In light of the challenges of MC estimates, we introduce a new way to get an unbiased estimate of the FIM. First, compute the scalar-valued function

$$\mathfrak{h}(\mathcal{D}_x, \theta) := \sum_{x \in \mathcal{D}_x} \sum_{y=1}^{C} \sqrt{\tilde{p}(y \mid x, \theta)} \ell_{xy}(\theta) \xi_{xy}, \tag{5}$$

where $\xi_{xy}$ is a standard multivariate Gaussian vector of size $C|\mathcal{D}|$ or a Rademacher vector, and $\tilde{p}(y \mid x, \theta)$ has the same value as $p(y \mid x, \theta)$ but is used with a *stop-gradient* operator, meaning its gradient is treated as zero, preventing error from backpropagating through $\tilde{p}(y \mid x, \theta)$. This $\tilde{p}$ can be implemented by `Tensor.detach()` in PyTorch (Paszke et al., 2019) or similar functions in other auto-differentiation (AD) frameworks. Second, the gradient vector $\frac{\partial \mathfrak{h}}{\partial \theta} = \sum_{x \in \mathcal{D}_x} \sum_{y=1}^{C} \sqrt{p(y \mid x, \theta)} \frac{\partial \ell_{xy}}{\partial \theta} \xi_{xy}$ can be evaluated via AD, e.g. by `h.backward()` in PyTorch. Third, the random psd matrix $\mathbb{F}(\theta) := \frac{\partial \mathfrak{h}}{\partial \theta} \frac{\partial \mathfrak{h}}{\partial \theta^{\top}}$, which we refer to as the "Hutchinson's estimate" (of the FIM), can be used to estimate $\mathcal{F}(\theta)$. By straightforward derivations,

$$\mathbb{E}_{p(\xi)} (\mathbb{F}(\theta)) = \sum_{x \in \mathcal{D}_x} \sum_{y=1}^{C} \sum_{x' \in \mathcal{D}_x} \sum_{y'=1}^{C} \sqrt{p(y \mid x, \theta)} \sqrt{p(y' \mid x', \theta)} \frac{\partial \ell_{xy}}{\partial \theta} \frac{\partial \ell_{x'y'}}{\partial \theta^{\top}} \mathbb{E}_{p(\xi)} [\xi_{xy} \xi_{x'y'}] = \mathcal{F}(\theta). \tag{6}$$

The last "=" is because $\mathbb{E}_{p(\xi)}(\xi_{xy}\xi_{x'y'}) = 1$ if $x = x'$ and $y = y'$, and $\mathbb{E}_{p(\xi)}(\xi_{xy}\xi_{x'y'}) = 0$ otherwise. Considering $\frac{\partial \mathfrak{h}}{\partial \theta}$ as an implicit representation of the FIM, its *computational cost* consists of ① evaluating the $\mathfrak{h}$ function; ② the backward pass to compute the gradient of $\mathfrak{h}$. The cost is the same as evaluating the gradient of the loss $-\sum_{x \in \mathcal{D}_x} \sum_{y=1}^{C} \ell_{xy}(\theta)$, noting that $\mathfrak{h}$ is the log-likelihood randomly flipped by a Gaussian/Rademacher vector. Moreover, $\mathfrak{h}$ can reuse the logits already computed during the forward pass. Therefore $\frac{\partial \mathfrak{h}}{\partial \theta}$ requires merely one additional backward pass, making it practical for large-scale networks. In summary, $\mathbb{F}(\theta)$ is a *model-agnostic, unbiased estimator* of $\mathcal{F}(\theta)$ for general statistical models, independent of neural network architectures and applicable to non-neural network models as well. Hutchinson's estimate has provable unbiasedness and variance bounds, as formally established below.

**Theorem 11.** *For all $\theta \in \Theta$, we have $\mathbb{E}_{p(\xi)}(\mathbb{F}(\theta)) = \mathcal{F}(\theta)$. If $p(\xi)$ is standard multivariate Gaussian, then $\mathrm{Var}(\mathbb{F}_{ii}(\theta)) = 2\mathcal{F}_{ii}(\theta)^2$; if $p(\xi)$ is standard multivariate Rademacher, then $\mathrm{Var}(\mathbb{F}_{ii}(\theta)) = 2\mathcal{F}_{ii}(\theta)^2 - 2\sum_{x \in \mathcal{D}_x} \sum_{y=1}^{C} p^2(y \,|\, x, \theta)(\frac{\partial \ell_{xy}}{\partial \theta_i})^4$.*

It is known that the Rademacher distribution yields smaller variance for Hutchinson's method compared to the Gaussian distribution. In what follows, $p(\xi)$ is Rademacher by default. By Theorem 11, $\mathrm{Std}(\mathbb{F}_{ii}(\theta)) \leq \sqrt{2}\mathcal{F}_{ii}(\theta)$. Thus the CV $\mathrm{Std}(\mathbb{F}_{ii}(\theta))/\mathcal{F}_{ii}(\theta)$ is bounded by $\sqrt{2}$. We only investigate the diagonal of Hutchinson's estimate because the diagonal FIM is widely used, but our results are readily extended to off-diagonal entries.

**Remark.** *For a dataset with $J$ minibatches, each with a diagonal FIM $\mathbb{F}_{ii}^{(j)}(\theta)$ computed with an independent probe, we have $\mathbb{F}_{ii}(\theta) = \sum_{j=1}^{J} \mathbb{F}_{ii}^{(j)}(\theta)$. By Theorem 11, $\mathrm{Var}(\mathbb{F}_{ii}(\theta)) = \sum_{j=1}^{J} \mathrm{Var}(\mathbb{F}_{ii}^{(j)}(\theta)) \leq 2\sum_{j=1}^{J}\left(\mathcal{F}_{ii}^{(j)}(\theta)\right)^2 \leq 2\left(\mathcal{F}_{ii}(\theta)\right)^2$. Moreover, we roughly approximate $\mathbb{F}_{ii}(\theta) \approx J\,\mathbb{F}_{ii}^{(j)}(\theta)$. Then, $\mathrm{Var}(\mathbb{F}_{ii}(\theta)) \leq 2\sum_{j=1}^{J}\left(\frac{\mathcal{F}_{ii}(\theta)}{J}\right)^2 = \frac{2}{J}\left(\mathcal{F}_{ii}(\theta)\right)^2$. At the dataset level, the variance is inversely proportional to $J$, while the computation cost grows linearly with $J$, presenting a typical accuracy–computation trade-off.*

**Remark.** *Taking trace on both sides of $\mathbb{E}_{p(\xi)}(\mathbb{F}(\theta)) = \mathcal{F}(\theta)$, we get $\mathbb{E}_{p(\xi)}(\|\frac{\partial \mathfrak{h}}{\partial \theta}\|^2) = \mathrm{tr}(\mathcal{F}(\theta))$. The squared Euclidean norm of $\frac{\partial \mathfrak{h}}{\partial \theta}$ is an unbiased estimate of the trace of the FIM. This is useful for efficient computations of related regularizers (Peebles et al., 2020).*

An alternative Hutchinson's estimate based on the equivalent FIM expression $\mathcal{F}(\theta) = 4\sum_{x \in \mathcal{D}_x} \sum_{y=1}^{C}\left[\frac{\partial\sqrt{p(y\,|\,x,\theta)}}{\partial\theta}\frac{\partial\sqrt{p(y\,|\,x,\theta)}}{\partial\theta^\top}\right]$ (see *e.g.* the first unnumbered equation in Sun & Nielsen 2017) is detailed in section B. We find that in practice its performance is similar to the above $\mathbb{F}$.

Note that a sample of the random matrix $\mathbb{F}(\theta)$ is always rank-1: $\mathrm{rank}\,\mathbb{F}(\theta) = 1 \leq \mathrm{rank}\,\mathcal{F}(\theta)$. Ideally, one can compute the numerical average of more than one $\mathbb{F}(\theta)$ samples to reduce variance and recover the rank, each requiring a separate backward pass. Due to computational constraints in deep learning practice, far fewer (*e.g.*, 1) samples are used. Instead, accumulated statistics along the learning path $\theta_1 \rightarrow \theta_2 \rightarrow \cdots$ can be used to maintain an (exponential) moving average of $\mathbb{F}(\theta_i)$. The underlying assumption is that $\theta_1, \theta_2, \cdots$ connected by small learning steps lie close to one another in the parameter space. Therefore, averaging $\mathbb{F}(\theta_i)$ provides a reasonable approximation of the local FIM with sufficient rank.

## 4.3 DIAGONAL CORE

For multi-label classification, and for computing the upper bound in Proposition 5, the core matrix is diagonal, in the form $\mathcal{I}^{\mathrm{DG}}(z(x,\theta)) = \mathrm{diag}\,(\zeta_1(x,\theta), \cdots, \zeta_C(x,\theta))$, and the associated FIM is $\mathcal{F}^{\mathrm{DG}}(\theta) = \sum_{x \in \mathcal{D}_x}\left(\frac{\partial z}{\partial \theta}\right)^\top \cdot \mathcal{I}^{\mathrm{DG}}(z(x,\theta)) \cdot \frac{\partial z}{\partial \theta}$. In the former case, $\zeta_y(x,\theta) = p(y\,|\,x,\theta)(1 - p(y\,|\,x,\theta))$; in the latter case, $\zeta_y(x,\theta) = p(y\,|\,x,\theta)$. Here, superscripts — *e.g.*, "DG" for diagonal and "LR(k)"/"LR" for low-rank — indicate the parametric form of the core FIM, in contrast to denoting the core space as in $\mathcal{I}^{\Delta}$. We define the scalar-valued function

$$\mathfrak{h}^{\mathrm{DG}}(\theta) := \sum_{x \in \mathcal{D}_x} \sum_{y=1}^{C} \sqrt{\tilde{\zeta}_y(x,\theta)}\,z_y(x,\theta)\xi_{xy}, \tag{7}$$

where $\xi_{xy}$ are standard Rademacher samples that are independent across all $x$ and $y$. Similar to the derivation steps in section 1, we first compute the random vector $\frac{\partial \mathfrak{h}^{\mathrm{DG}}}{\partial \theta}$ through AD, and then compute $\mathbb{F}^{\mathrm{DG}}(\theta) := \frac{\partial \mathfrak{h}^{\mathrm{DG}}}{\partial \theta} \frac{\partial \mathfrak{h}^{\mathrm{DG}}}{\partial \theta^{\top}}$ (or its diagonal blocks) to estimate $\mathcal{F}^{\mathrm{DG}}(\theta)$.

For computing the upper bound in Proposition 5, $\tilde{\zeta}_y(x, \theta) = \tilde{p}_y(x, \theta)$, then we find that Eq. (5) and Eq. (7) are similar. The only difference is that, the "raw" logits $z_y$ in Eq. (7) are replaced by $\ell_{xy}(\theta) = z_y - \log \sum_y \exp(z_y)$ in Eq. (5). Compared to $\frac{\partial z_y}{\partial \theta}$, the gradient $\frac{\partial \ell_{xy}}{\partial \theta} = \frac{\partial z_y}{\partial \theta} - \sum_y p(y \mid x, \theta) \frac{\partial z_y}{\partial \theta}$ is centered. Due to their computational similarity, in practice, one should use Eq. (5) instead of Eq. (7) and get an unbiased estimate of $\mathcal{F}^{\Delta}(\theta)$. Eq. (7) is useful when the dimensions of $y$ are conditionally independent given $x$, e.g. for computing $\mathcal{F}^{\mathcal{C}}(\theta)$.

## 4.4 LOW-RANK CORE

By Proposition 5, $\mathcal{F}^{\Delta}(\theta) \succeq \mathcal{F}^{\mathrm{LR(k)}}(\theta) := \sum_{x \in \mathcal{D}_x} \sum_{i=C-k+1}^{C} \lambda_i(x, \theta) \left(\frac{\partial z}{\partial \theta}\right)^{\top} v_i(x, \theta) v_i^{\top}(x, \theta) \frac{\partial z}{\partial \theta}$. We define

$$\mathfrak{h}^{\mathrm{LR(k)}}(\theta) = \sum_{x \in \mathcal{D}_x} \sum_{i=C-k+1}^{C} \sqrt{\tilde{\lambda}_i(x, \theta)} \tilde{v}_i^{\top}(x, \theta) z(x, \theta) \xi_{xi}, \tag{8}$$

where $\xi_{xi}$ are independent standard Rademacher samples, and $k \in \{1, \cdots, C-1\}$. For computing $\mathfrak{h}^{\mathrm{LR(k)}}(\theta)$, we only need $k|\mathcal{D}_x|$ Rademacher samples, as compared to $C|\mathcal{D}_x|$ samples for computing $\mathfrak{h}(\theta)$ and $\mathfrak{h}^{\mathrm{DG}}(\theta)$. Correspondingly, $\mathbb{F}^{\mathrm{LR(k)}}(\theta) := \frac{\partial \mathfrak{h}^{\mathrm{LR(k)}}}{\partial \theta} \frac{\partial \mathfrak{h}^{\mathrm{LR(k)}}}{\partial \theta^{\top}}$ is used to estimate $\mathcal{F}^{\mathrm{LR(k)}}(\theta)$. When $k = 1$, we simply denote $\mathcal{F}^{\mathrm{LR}} := \mathcal{F}^{\mathrm{LR(1)}}$, $\mathfrak{h}^{\mathrm{LR}} := \mathfrak{h}^{\mathrm{LR(1)}}$, and $\mathbb{F}^{\mathrm{LR}} := \mathbb{F}^{\mathrm{LR(1)}}$.

It remains to compute $\lambda_i(x, \theta)$ and $v_i(x, \theta)$, which requires a spectral decomposition of a $C \times C$ matrix for each $x \in \mathcal{D}_x$. The cost is only acceptable when $C$ is small to moderate. In our CIFAR-100 experiments ($C = 100$), the computational speed of $\mathbb{F}^{\mathrm{LR(k)}}$ drops to roughly half that of $\mathbb{F}$. If $k = 1$, however, $\lambda_C(x, \theta)$ and $v_C(x, \theta)$ can be computed more efficiently using the power iteration. By Eq. (3), starting from a random unit vector $v_C^0$, we compute

$$v_C^{t+1} = \frac{\mathcal{I}^{\Delta}(z) v_C^t}{\|\mathcal{I}^{\Delta}(z) v_C^t\|} = \frac{p \circ v_C^t - p^{\top} v_C^t p}{\|p \circ v_C^t - p^{\top} v_C^t p\|},$$

for $t = 1, 2, \cdots$, until convergence or a fixed number of iterations is reached. Then, $\lambda_C = p^{\top}(v_C \circ v_C) - (p^{\top} v_C)^2$. For computing $\lambda_C$ and $v_C$ for all $x \in \mathcal{D}_x$, the per-iteration computational cost is $\mathcal{O}(C|\mathcal{D}_x|)$, so the total cost is $\mathcal{O}(TC|\mathcal{D}_x|)$ for $T$ iterations. The number of iterations required increases as the spectral gap $\gamma := \lambda_C - \lambda_{C-1}$ decreases. Convergence can be slow when $\gamma$ is small (e.g., for near-uniform output distributions). In our implementation, we simply use a fixed iteration budget (e.g., $T = 30$). All our estimators: $\mathfrak{h}$, $\mathfrak{h}^{\mathrm{DG}}$ and $\mathfrak{h}^{\mathrm{LR(k)}}$ can be computed solely based on the neural network output logits $z(x, \theta)$ for each $x \in \mathcal{D}_x$.

## 4.5 NUMERICAL SIMULATIONS

We compute the *diagonal FIM* of the following models: ① DistilBERT (Sanh et al., 2019; Wolf et al., 2020) fine-tuned on the Stanford Sentiment Treebank v2 (SST-2) (Socher et al., 2013) (with $C = 2$ classes); ② DistilBERT (pretrained) with a randomly initialized classification head for DBpedia ontology classification (Lehmann et al., 2015) ($C = 14$); ③ RoBERTa-base (Liu et al., 2019) fine-tuned on Multi-Genre Natural Language Inference (MNLI) corpus (Williams et al., 2018) ($C = 3$); ④ ImageNet-pretrained ResNet-50 (He et al., 2016) with a random classification head for CIFAR-100 image classification (Krizhevsky, 2009) ($C = 100$); ⑤ Same as ④ but with an ImageNet-pretrained EfficientNet-B0 (Tan & Le, 2019) backbone; ⑥ Wav2Vec2-base (Baevski et al., 2020) (pretrained) with a random classification head on SpeechCommands audio classification (Warden, 2018) ($C = 12$).

For all datasets, the diagonal FIM $\mathcal{F}_{ii}$ is computed on a fixed random subset of 128 batches with a batch size of $B = 64$. We evaluate the ground-truth $\mathcal{F}_{ii}$ using its closed-form expression in Eq. (1), which requires $8192C$ backward passes and is impractical to use on the full dataset. Figure 1 shows the histograms of $\mathcal{F}_{ii}$ on RoBERTa-base, ResNet-50, and Wav2Vec2-base, including the zero atom (probability mass at zero). Other datasets and models are omitted due to space constraints. The

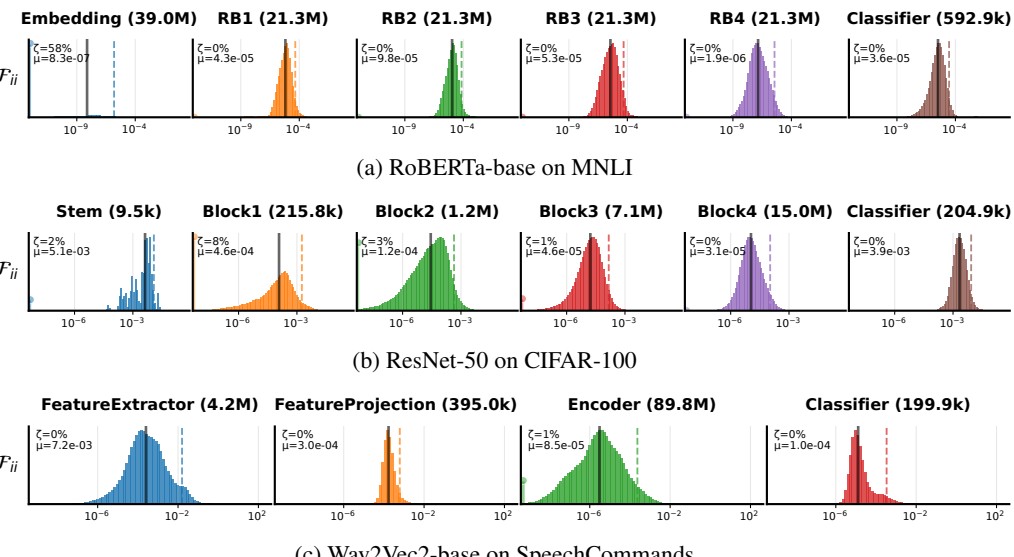

Figure 1: Histograms of the ground-truth diagonal FIM entries $\mathcal{F}_{ii}$ on a logarithmic x-axis. The zero atom is displayed as a vertical bar at the left edge of each plot. From top to bottom, three different models (associated with three different neuromanifolds) are shown. From left to right, successive components from input to output and their parameter counts are displayed. $\zeta$ denotes the zero probability. $\mu$ denotes the average value of $\mathcal{F}_{ii}$ in the component. The solid and dashed vertical lines indicate the median and the $p_{95}$ quantile of strictly positive values, respectively.

distribution of $\mathcal{F}_{ii}$ differs substantially across tasks. For example, in RoBERTa-base, the embedding layers exhibit a large atom at zero corresponding to unobserved vocabulary, whereas intermediate transformer layers show the largest Fisher information. Similar patterns are observed in other NLP tasks.

We only compare FIM estimators that can be computed using *a single backward pass* per batch, including the empirical FIM $\overline{\mathcal{F}}_{ii}(\theta)$, $\mathbb{F}_{ii}(\theta)$ (Hutchinson's unbiased estimate), $\mathbb{F}_{ii}^{\mathrm{DG}}(\theta)$ (upper-biased estimate of $\mathcal{F}_{ii}$), $\mathbb{F}_{ii}^{\mathrm{LR}}(\theta)$, and $\mathbb{F}_{ii}^{\mathrm{LR}(2)}(\theta)$ (lower-biased estimate of $\mathcal{F}_{ii}$). The MC estimate $\hat{\mathcal{F}}$ is excluded, because it requires $B$ backward passes per batch ($B$: batch size) and is less applicable to production settings. Table 2 shows the *relative mean absolute error* (RelMAE), defined as the average ratio of the absolute error to the ground-truth value, with $\varepsilon = 10^{-12}$ added for numerical stability. For example, the RelMAE of empirical FIM is $\frac{1}{\dim(\theta)} \sum_{i=1}^{\dim(\theta)} \frac{|\overline{\mathcal{F}}_{ii} - \mathcal{F}_{ii}|}{\mathcal{F}_{ii} + \varepsilon}$. Because $\mathcal{F}_{ii}$ is typically small in magnitude, RelMAE is more interpretable than the mean absolute error (MAE). In general, $\mathbb{F}_{ii}$ is the most accurate, with a RelMAE of approximately 0.2, corresponding to $\pm 20\%$ relative deviation from the ground truth. This improvement arises because $\mathbb{F}$ is unbiased, whereas other baselines are biased. Nevertheless, $\mathcal{F}_{ii}^{\mathrm{LR}}$ and $\mathcal{F}_{ii}^{\mathrm{LR}(2)}$ are the most accurate on SST-2 and MNLI. This is because, on these two tasks, the model is fine-tuned and the core FIM exhibits an approximately low-rank structure. The empirical FIM and $\mathbb{F}_{ii}^{\mathrm{DG}}$ are the least accurate.

The computational speeds of all methods are broadly similar. Hutchinson's estimate is as fast as the empirical FIM. In contrast, $\mathcal{F}_{ii}^{\mathrm{LR}}$ and $\mathcal{F}_{ii}^{\mathrm{LR}(2)}$ are more expensive, because they rely on power iterations or spectral decompositions of the core FIM. The bottom line is: to compute the diagonal FIM, we recommend Hutchinson's unbiased estimate $\mathbb{F}$ over the empirical FIM $\overline{\mathcal{F}}$. For fine-tuned models, one may alternatively use $\mathbb{F}^{\mathrm{LR}}$ or $\mathbb{F}^{\mathrm{LR}(k)}$ for potentially improved accuracy.

# 5 RELATED WORK

A prominent application of Fisher information in deep learning is the natural gradient (Amari, 1998) and its variants. The Adam optimizer (Kingma & Ba, 2015) uses the empirical diagonal FIM.

Table 2: RelMAE w.r.t. the ground-truth diagonal FIM entries $\mathcal{F}_{ii}$ for different FIM estimators (columns) across tasks (rows). Numbers in parentheses mean speedup factors relative to the empirical FIM (larger is faster). CIFAR-100 is used for both ResNet-50 (R) and EfficientNet-B0 (E).

| | $\overline{\mathcal{F}}_{ii}$ | $\mathbb{F}_{ii}$ | $\mathbb{F}_{ii}^{\mathrm{DG}}$ | $\mathbb{F}_{ii}^{\mathrm{LR}}$ | $\mathbb{F}_{ii}^{\mathrm{LR}(2)}$ |
|---|---|---|---|---|---|
| SST-2 | $1.15\,(\times 1)$ | $0.18\,(\times 1.07)$ | $341\,(\times 1.07)$ | $\mathbf{0.05}\,(\times 0.96)$ | $\mathbf{0.05}\,(\times 1.00)$ |
| DBpedia | $0.59\,(\times 1)$ | $\mathbf{0.22}\,(\times 1.00)$ | $0.25\,(\times 1.00)$ | $0.8\,(\times 0.93)$ | $0.72\,(\times 0.93)$ |
| MNLI | $53.9\,(\times 1)$ | $0.16\,(\times 1.00)$ | $8.36\,(\times 0.97)$ | $\mathbf{0.11}\,(\times 0.96)$ | $0.12\,(\times 0.95)$ |
| CIFAR-100 (R) | $0.17\,(\times 1)$ | $\mathbf{0.11}\,(\times 0.99)$ | $\mathbf{0.11}\,(\times 1.01)$ | $0.97\,(\times 0.97)$ | $0.95\,(\times 0.46)$ |
| CIFAR-100 (E) | $0.17\,(\times 1)$ | $\mathbf{0.11}\,(\times 1.00)$ | $0.12\,(\times 1.00)$ | $0.98\,(\times 0.98)$ | $0.96\,(\times 0.50)$ |
| SpeechCommands | $56.8\,(\times 1)$ | $\mathbf{0.17}\,(\times 0.97)$ | $7.4\,(\times 0.97)$ | $0.39\,(\times 0.89)$ | $0.22\,(\times 0.91)$ |

Efforts have been made to obtain more accurate approximations of $\mathcal{F}(\theta)$ at the expense of higher computational cost, such as modeling the diagonal blocks of $\mathcal{F}(\theta)$ with Kronecker products (Martens, 2020) or component-wise FIMs (Ollivier, 2015; Sun & Nielsen, 2017), or computing $\mathcal{F}(\theta)$ through low-rank approximations (Le Roux et al., 2007; Botev et al., 2017). The FIM can be alternatively defined on a sub-model (Sun & Nielsen, 2017) instead of the global mapping $x \rightarrow y$ or based on $\alpha$-embeddings of a parametric family (Nielsen, 2017). AdaHessian (Yao et al., 2021) uses Hutchinson probes to approximate the diagonal Hessian.

From theoretical perspectives, the quality of the Kronecker approximation is discussed (Martens & Grosse, 2015) with its error bounded. It is well known that the eFIM differs from $\mathcal{F}(\theta)$ (Pascanu & Bengio, 2014; Martens, 2020; Kunstner et al., 2020) and leads to distinct optimization paths. The accuracy of two different MC approximations of $\mathcal{F}(\theta)$ is analyzed (Guo & Spall, 2019; Soen & Sun, 2021; 2024; Sun & Spall, 2021), which lie in the framework of MC information geometry (Nielsen & Hadjeres, 2019). In our analysis, the Hutchinson estimate $\mathbb{F}(\theta)$ has unique advantages over both MC and the eFIM. Notably, the MC estimate requires computing $\frac{\partial \ell_{\hat{x}_k \hat{y}_k}}{\partial \theta}$ for multiple samples of $(\hat{x}_k, \hat{y}_k)$, while $\mathbb{F}(\theta)$ typically needs to evaluate one gradient vector $\frac{\partial \mathfrak{h}}{\partial \theta}$. Our bounds improve over existing bounds, *e.g.* those of $\mathcal{F}(\theta)$ (Soen & Sun, 2024), through carefully analyzing the core space.

Hutchinson's stochastic trace estimator is used to estimate the trace of the FIM (Jastrzebski et al., 2021), or the FIM for Gaussian processes (Stein et al., 2013; Geoga et al., 2020) where the FIM entries are in the form of a trace. Closely related to this are computations involving the Hessian, where Hutchinson's trick is applied to compute the Hessian trace (Hu et al., 2024), the principal curvature (Böttcher & Wheeler, 2024), or related regularizers (Peebles et al., 2020). The Hessian trace estimator is implemented in deep learning libraries (Dangel et al., 2020; Yao et al., 2020) and usually relies on the Hessian-vector product. Our estimator $\mathbb{F}$ differs from prior efforts in three key aspects: ① unbiased for the true FIM; ② coming with explicit variance bounds; ③ scalable to modern architectures. Crucially, no previous method simultaneously satisfies all three properties. By leveraging both Hutchinson's trick and AD interfaces, $\mathbb{F}$ avoids the need for expensive Hessian computations or approximations, and is well-suited in scalable settings. As demonstrated in the experiments, $\mathbb{F}$ can be applied to different classifier networks regardless of the architecture.

## 6 CONCLUSION

We explore the FIM $\mathcal{F}(\theta)$ of classifier networks, focusing on the case of multi-class classification. We provide deterministic lower and upper bounds of the FIM based on related bounds in the low-dimensional core space. We discover a new family of random estimators $\mathbb{F}(\theta)$ based on Hutchinson's method. They offer theoretical guarantees on estimation quality (bounded variance) and can be computed efficiently via auto-differentiation. The proposed $\mathbb{F}(\theta)$ is readily integrated into deep learning libraries (Dangel et al., 2020; Yao et al., 2020) for efficiently evaluating the FIM or approximating the Hessian. Our analysis in the core space gives insights and useful tools for information geometry where the probability simplex is widely used. As a limitation, the results here concern static computations of $\mathcal{F}(\theta)$ at a fixed $\theta$ but are not directly integrated into downstream applications that employ $\mathbb{F}(\theta)$ during learning. For example, new deep learning optimizers are not developed here and left as future work. Advanced variance-reduction techniques (Meyer et al., 2021) that could improve $\mathbb{F}(\theta)$ remain to be investigated.

ACKNOWLEDGMENTS

The author thanks Frank Nielsen and the anonymous reviewers for their insightful feedback.

ETHICS STATEMENT

The author has read the ICLR Code of Ethics and confirms that this research fully complies with the Code of Ethics.

REPRODUCIBILITY STATEMENT

All assumptions and proofs of the formal statements are provided in the main text and the appendix. A PyTorch library to compute Hutchinson's FIM for general deep classifiers is available at https://github.com/sunk/fim-estimators

THE USE OF LARGE LANGUAGE MODELS (LLMS)

LLMs are used as a side reference for language editing purpose (grammar, wording, and translation). LLMs are not used in developing the technical contents or the core results.

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

## A   FURTHER ANALYSIS IN THE CORE SPACE

The lemma below gives the average error (variance) of using $R(y)$ to estimate $\mathcal{I}^\Delta(z)$, where $y$ is a random variable distributed according to $p(y \mid z)$.

**Lemma 12.** *The element-wise variance of the random matrix $R(y)$, denoted by $\mathrm{Var}(R_{ij})$, is given by*

$$\mathrm{Var}(R_{ij}) = \begin{cases} p_i(1-p_i)(1-4p_i(1-p_i)) & \text{if } i = j; \\ p_i p_j (p_i + p_j - 4p_i p_j) & \text{otherwise.} \end{cases}$$

$\forall i, j, \ \mathrm{Var}(R_{ij}) \le 1/16$. *For both diagonal and off-diagonal entries, the coefficient of variation (CV)* $\mathrm{Std}(R_{ij})/|\mathcal{I}^\Delta_{ij}(z)|$ *can be arbitrarily large, where* $\mathrm{Std}(\cdot)$ *means standard deviation.*

*Proof.* We first look at the diagonal entries of $R(y)$. We have

$$R_{ii} = (\llbracket y = i \rrbracket - p_i)^2 = \begin{cases} (1-p_i)^2 & \text{if } y = i; \\ p_i^2 & \text{otherwise.} \end{cases}$$

Therefore
$$\mathbb{E}(R_{ii}) = p_i(1 - p_i)^2 + (1 - p_i)p_i^2 = p_i(1 - p_i) = \mathcal{I}_{ii}^{\Delta}(z).$$
This shows that $R_{ii}$ is an unbiased estimator of the diagonal entries of $\mathcal{I}^{\Delta}(z)$. We have
$$\mathbb{E}(R_{ii}^2) = p_i(1 - p_i)^4 + (1 - p_i)p_i^4 = p_i(1 - p_i)\left[(1 - p_i)^3 + p_i^3\right]$$
$$= p_i(1 - p_i)\left[(1 - p_i)^2 - p_i(1 - p_i) + p_i^2\right].$$
Therefore
$$\begin{aligned}
\mathrm{Var}(R_{ii}) &= \mathbb{E}(R_{ii}^2) - (\mathbb{E}(R_{ii}))^2 \\
&= p_i(1 - p_i)\left[(1 - p_i)^2 - p_i(1 - p_i) + p_i^2\right] - p_i^2(1 - p_i)^2 \\
&= p_i(1 - p_i)\left[(1 - p_i)^2 - 2p_i(1 - p_i) + p_i^2\right] \\
&= p_i(1 - p_i)(1 - 4p_i(1 - p_i)) \\
&= \mathcal{I}_{ii}^{\Delta}(z)(1 - 4\mathcal{I}_{ii}^{\Delta}(z)) \\
&= -4\left(\mathcal{I}_{ii}^{\Delta}(z) - \frac{1}{8}\right)^2 + \frac{1}{16} \leq \frac{1}{16}.
\end{aligned}$$
The coefficient of variation (CV)
$$\frac{\sqrt{\mathrm{Var}(R_{ii})}}{\mathcal{I}_{ii}^{\Delta}(z)} = \sqrt{\frac{\mathcal{I}_{ii}^{\Delta}(z)(1 - 4\mathcal{I}_{ii}^{\Delta}(z))}{\mathcal{I}_{ii}^{\Delta}(z)^2}} = \sqrt{\frac{1}{\mathcal{I}_{ii}^{\Delta}(z)} - 4}$$
is unbounded. As $\mathcal{I}_{ii}^{\Delta}(z) \to 0$, the CV can take arbitrarily large values.

Next, we consider the off-diagonal entries of $R$. For $i \neq j$, we have
$$\begin{aligned}
R_{ij} &= (\llbracket y = i \rrbracket - p_i)(\llbracket y = j \rrbracket - p_j) \\
&= p_i p_j - \llbracket y = i \rrbracket p_j - \llbracket y = j \rrbracket p_i.
\end{aligned}$$
Hence,
$$\mathbb{E}(R_{ij}) = p_i p_j - p_i p_j - p_j p_i = -p_i p_j = \mathcal{I}_{ij}^{\Delta}(z).$$
At the same time,
$$\begin{aligned}
\mathbb{E}(R_{ij}^2) &= \mathbb{E}\left(p_i p_j - \llbracket y = i \rrbracket p_j - \llbracket y = j \rrbracket p_i\right)^2 \\
&= p_i^2 p_j^2 + \mathbb{E}\left(\llbracket y = i \rrbracket p_j^2 + \llbracket y = j \rrbracket p_i^2 - 2\llbracket y = i \rrbracket p_i p_j^2 - 2\llbracket y = j \rrbracket p_i^2 p_j\right) \\
&= p_i^2 p_j^2 + p_i p_j^2 + p_j p_i^2 - 2p_i^2 p_j^2 - 2p_i^2 p_j^2 \\
&= p_i p_j^2 + p_i^2 p_j - 3p_i^2 p_j^2 \\
&= p_i p_j(p_i + p_j - 3p_i p_j).
\end{aligned}$$
Therefore
$$\begin{aligned}
\mathrm{Var}(R_{ij}) &= \mathbb{E}(R_{ij}^2) - (\mathbb{E}(R_{ij}))^2 \\
&= p_i p_j(p_i + p_j - 3p_i p_j) - p_i^2 p_j^2 \\
&= p_i p_j(p_i + p_j - 4p_i p_j) \\
&\leq p_i p_j(1 - 4p_i p_j) \\
&= -4\left(p_i p_j - \frac{1}{8}\right)^2 + \frac{1}{16} \leq \frac{1}{16}.
\end{aligned}$$
The coefficient of variation
$$\frac{\sqrt{\mathrm{Var}(R_{ij})}}{|\mathcal{I}_{ij}^{\Delta}(z)|} = \sqrt{\frac{p_i p_j(p_i + p_j - 4p_i p_j)}{p_i^2 p_j^2}} = \sqrt{\frac{1}{p_i} + \frac{1}{p_j} - 4}$$
is unbounded. As either $p_i \to 0$ or $p_j \to 0$, the CV can take arbitrarily large values. $\qquad\square$

By Lemma 12, when using the rank-1 matrix $R(y)$ as an estimator of $\mathcal{I}^{\Delta}(z)$, the absolute error is bounded, but the relative error given by the CV is unbounded. One may alternatively use the rank-2 random matrix $R'(y) = e_{yy} - pp^{\top}$ to estimate $\mathcal{I}^{\Delta}(z)$. Obviously we have $\mathbb{E}(R'(y)) = \mathrm{diag}\,(p) - pp^{\top} = \mathcal{I}^{\Delta}(z)$ and thus $R'(y)$ is unbiased. The variance appears only on the diagonal, while all off-diagonal entries are deterministic with zero variance. This $R'(y)$ is not used in our development but is of theoretical interest.

## B   An Alternative Hutchinson Estimator

We can rewrite the FIM in Eq. (1) as

$$\mathcal{F}(\theta) = 4 \sum_{x \in \mathcal{D}_x} \sum_{y=1}^{C} \left[ \frac{\partial \sqrt{p(y \mid x, \theta)}}{\partial \theta} \frac{\partial \sqrt{p(y \mid x, \theta)}}{\partial \theta^\top} \right].$$

We define

$$\mathfrak{h}^{\mathrm{sqrt}}(\mathcal{D}_x, \theta) = 2 \sum_{x \in \mathcal{D}_x} \sum_{y=1}^{C} \sqrt{p(y \mid x, \theta)} \xi_{xy}, \tag{9}$$

where $\{\xi_{xy}\}$ are i.i.d. standard normal (or Rademacher) random variables. Then, we can use AD to compute

$$\frac{\partial \mathfrak{h}^{\mathrm{sqrt}}}{\partial \theta} = 2 \sum_{x \in \mathcal{D}_x} \sum_{y=1}^{C} \frac{\partial \sqrt{p(y \mid x, \theta)}}{\partial \theta} \xi_{xy},$$

Then,

$$\mathbb{F}^{\mathrm{sqrt}}(\theta) := \frac{\partial \mathfrak{h}^{\mathrm{sqrt}}}{\partial \theta} \frac{\partial \mathfrak{h}^{\mathrm{sqrt}}}{\partial \theta^\top} \tag{10}$$

gives an unbiased estimate of the FIM $\mathcal{F}(\theta)$, with bounded variance (details are straightforward and omitted for brevity).

This $\mathbb{F}^{\mathrm{sqrt}}$ differs from $\mathbb{F}$ in two aspects:

- It requires no `detach()` operation;
- The square root can be avoided by noting

$$\sqrt{p(y \mid x, \theta)} = \exp \left( \frac{1}{2} \left( z_y(x, \theta) - \log \sum_y \exp(z_y(x, \theta)) \right) \right),$$

where $z_y(x, \theta) - \log \sum_y \exp(z_y(x, \theta))$ can be computed via PyTorch's `log_softmax()` method.

$\mathbb{F}^{\mathrm{sqrt}}$ is numerically more stable because it does not require clipping the operand inside the square root to be above zero. In our experiments, however, we notice little difference compared to $\mathbb{F}$. All presented experimental results are produced using $\mathbb{F}$ introduced in the main text.

## C   Supplementary Experiments

Figure 2 shows the distribution of the ground-truth diagonal FIMs of DistilBERT on SST-2, Distil-BERT on DBpedia, and EfficientNet-B0 on CIFAR-100. The classification head exhibits the largest Fisher information among all components at random initialization, whereas its Fisher information is comparatively small in fine-tuned models. In an early draft, we included experiments on DistilBERT for AG News (Zhang et al., 2015) topic classification ($C = 4$ classes), which has been streamlined to make space for other types of datasets and to present a more representative range of class counts $C$. All numerical results presented in this paper were obtained on NVIDIA H100 SXM5 GPUs.

## D   Proof of Theorem 1

*Proof.* Recall the closed-form expression for the simplex FIM:

$$\mathcal{I}^\Delta(z) = \operatorname{diag}(p) - pp^\top.$$

Therefore

$$\mathcal{I}^\Delta(z)e = (\operatorname{diag}(p) - pp^\top)e = p - \left( \sum_{i=1}^{C} p_i \right) p = p - p = 0.$$

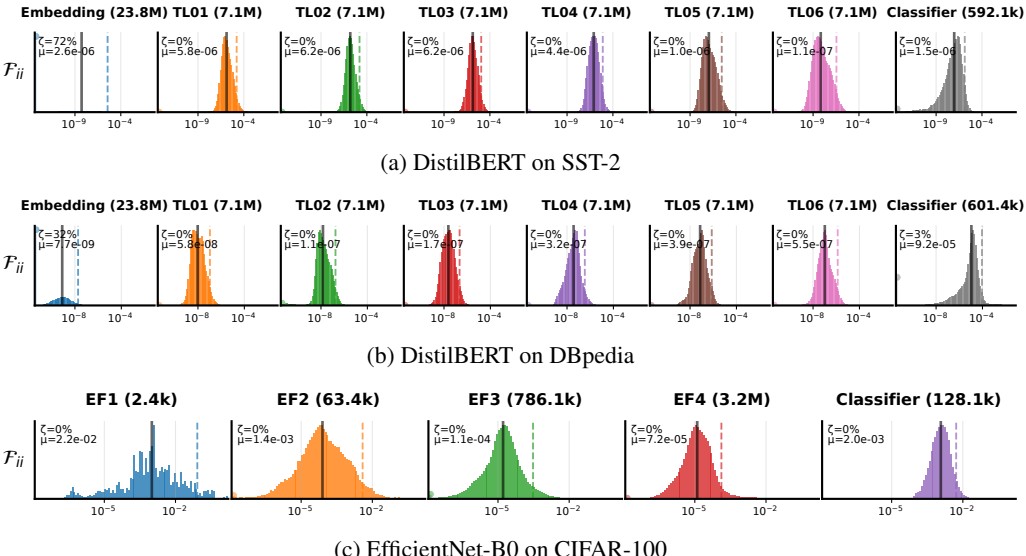

Figure 2: Histograms of the ground-truth diagonal FIM entries $\mathcal{F}_{ii}$ on a logarithmic x-axis. The zero atom is displayed as a vertical bar at the left edge of each plot. From left to right, successive components from input to output and their parameter counts are displayed. $\zeta$ denotes the zero probability. $\mu$ denotes the average value of $\mathcal{F}_{ii}$ in the component. The solid and dashed vertical lines indicate the median and the $p_{95}$ quantile of strictly positive values, respectively.

Therefore $\{te \ : \ t \in \Re\}$ is the one-dimensional kernel of $\mathcal{I}^\Delta(z)$. Since $\mathcal{I}^\Delta(z) \succeq 0$, we must have $\lambda_1 = 0$, and $v_1 = e/\|e\|$.

To analyze the sum of the eigenvalues of $\mathcal{I}^\Delta(z)$, we have

$$\sum_{i=1}^{C} \lambda_i = \text{tr}(\mathcal{I}^\Delta(z)) = \text{tr}(\text{diag}\,(p)) - \text{tr}(pp^\top) = 1 - \text{tr}(p^\top p) = 1 - p^\top p = 1 - \|p\|^2.$$

Below, we consider the maximum eigenvalue $\lambda_C$. We know that

$$\lambda_C = \sup_{\|u\|=1} u^\top \mathcal{I}^\Delta(z)u.$$

Therefore

$$\forall i, \quad \lambda_C \geq e_i^\top \mathcal{I}^\Delta(z)e_i = \mathcal{I}_{ii}^\Delta(z) = p_i(1 - p_i).$$

Therefore $\lambda_C \geq \max_i p_i(1 - p_i)$. At the same time, because $\lambda_1 = 0$, we have

$$\sum_{i=1}^{C} \lambda_i = \lambda_2 + \lambda_3 + \cdots + \lambda_C \leq (C - 1)\lambda_C.$$

Therefore

$$\lambda_C \geq \frac{\sum_{i=1}^{C} \lambda_i}{C - 1} = \frac{1 - \|p\|^2}{C - 1}.$$

Because

$$\text{diag}\,(p) = \mathcal{I}^\Delta(z) + pp^\top.$$

By Cauchy's interlacing theorem, we have

$$\lambda_{C-1} \leq p_{(C-1)} \leq \lambda_C \leq p_{(C)}.$$

It remains to prove the upper bounds of $\lambda_C$. First, we have

$$\lambda_C = \sup_{\|u\|=1} u^\top \mathcal{I}^\Delta(z)u. = \sup_{\|u\|=1} \left( \sum_{i=1}^{C} p_i u_i^2 - (p^\top u)^2 \right)$$

$$\leq \sup_{\|u\|=1} \sum_{i=1}^{C} p_i u_i^2 = \max_i p_i = p_{(C)},$$

which has just been proved using Cauchy's interlacing theorem.

By Gershgorin circle theorem, $\lambda_C$ must lie in one of the Gershgorin discs, given by the closed intervals

$$\left[ p_i(1-p_i) - \sum_{j\neq i} p_i p_j, \ p_i(1-p_i) + \sum_{j\neq i} p_i p_j \right], \quad i = 1, \cdots, C.$$

Therefore

$$\lambda_C \leq \max_i \left( p_i(1-p_i) + \sum_{j\neq i} p_i p_j \right)$$

$$= \max_i \left( p_i(1-p_i) + p_i(1-p_i) \right) = 2 \max_i p_i(1-p_i).$$

Because $\mathcal{I}^\Delta(z) \succeq 0$,

$$\lambda_C \leq \sum_{i=1}^{C} \lambda_i = 1 - \|p\|^2.$$

The statement follows immediately by combining the above lower and upper bounds of $\lambda_C$. $\qquad\square$

## E   PROOF OF LEMMA 2

*Proof.* Because $\mathcal{I}^\Delta(z) \succeq 0$, all its eigenvalues are non-negative. We have

$$\mathcal{I}^\Delta(z) - \lambda_C v_C v_C^\top = \sum_{i=1}^{C-1} \lambda_i v_i v_i^\top \succeq 0.$$

We show that $\lambda_C v_C v_C^\top$ is the best rank-1 representation of $\mathcal{I}^\Delta(z)$. Assume there exists $u \neq 0$ such that $\mathcal{I}^\Delta(z) \succeq uu^\top \succeq \lambda_C v_C v_C^\top$. Then

$$v_C^\top \mathcal{I}^\Delta(z) v_C = \lambda_C \geq (v_C^\top u)^2 \geq \lambda_C.$$

Therefore

$$v_C^\top u = \pm\sqrt{\lambda_C}.$$

Assume that $u = \sum_{i=1}^{C} \alpha_i v_i$, then $\alpha_C = v_C^\top u = \pm\sqrt{\lambda_C}$. Moreover, we have

$$\lambda_C \geq \frac{u^\top}{\|u\|} \mathcal{I}^\Delta(z) \frac{u}{\|u\|} \geq \frac{u^\top}{\|u\|} uu^\top \frac{u}{\|u\|} = \|u\|^2 = \sum_{i=1}^{C} \alpha_i^2.$$

Therefore, $\forall i \neq C$, $\alpha_i = 0$. In summary, $u = \pm\sqrt{\lambda_C} v_C$. Hence, $uu^\top = \lambda_C v_C v_C^\top$.

We have

$$\mathrm{diag}\,(p) - \mathcal{I}^\Delta(z) = \mathrm{diag}\,(p) - (\mathrm{diag}\,(p) - pp^\top) = pp^\top \succeq 0.$$

Therefore $\mathrm{diag}\,(p) \succeq \mathcal{I}^\Delta(z)$. Assume that $\mathrm{diag}\,(q)$ satisfies

$$\mathcal{I}^\Delta(z) \preceq \mathrm{diag}\,(q) \preceq \mathrm{diag}\,(p).$$

Then

$$\mathrm{diag}\,(p) - \mathcal{I}^\Delta(z) = pp^\top \succeq \mathrm{diag}\,(q) - \mathcal{I}^\Delta(z) \succeq 0.$$

Therefore

$$\mathrm{diag}\,(q) - \mathcal{I}^\Delta(z) = \beta pp^\top \ (\beta \leq 1).$$

Consequently,

$$\mathrm{diag}\,(q) = \mathcal{I}^\Delta(z) + \beta pp^\top = \mathrm{diag}\,(p) - pp^\top + \beta pp^\top = \mathrm{diag}\,(p) + (\beta - 1)pp^\top.$$

Therefore all off-diagonal entries of $(\beta - 1)pp^\top$ are zero. We must have $\beta = 1$ and thus $\mathrm{diag}\,(q) = \mathrm{diag}\,(p)$. $\qquad\square$

## F    PROOF OF LEMMA 3

*Proof.* We have

$$\|\lambda_C v_C v_C^\top - \mathcal{I}^\Delta(z)\| = \|\sum_{i=1}^{C-1} \lambda_i v_i v_i^\top\| = \sqrt{\sum_{i=1}^{C-1} \lambda_i^2} \leq \sqrt{(\sum_{i=1}^{C-1} \lambda_i)^2}$$

$$= \sum_{i=1}^{C-1} \lambda_i = \operatorname{tr}(\mathcal{I}^\Delta(z)) - \lambda_C = 1 - \|p\|^2 - \lambda_C.$$

By Theorem 1, we have $\lambda_C \geq p_{(C-1)}$. Therefore

$$\|\lambda_C v_C v_C^\top - \mathcal{I}^\Delta(z)\| \leq 1 - \|p\|^2 - p_{(C-1)}.$$

By Cauchy's interlacing theorem (see our proof of Theorem 1), we have

$$\forall i \in \{1, \cdots, C-1\}, \quad \lambda_i \leq p_{(i)}.$$

Hence

$$\|\lambda_C v_C v_C^\top - \mathcal{I}^\Delta(z)\| = \sqrt{\sum_{i=1}^{C-1} \lambda_i^2} = \sqrt{\sum_{i=2}^{C-1} \lambda_i^2} \leq \sqrt{\sum_{i=2}^{C-1} p_{(i)}^2}.$$

The statement follows immediately by combining the above upper bounds. $\square$

## G    PROOF OF LEMMA 4

*Proof.* The spectrum of $R(y)$ is

$$0 \leq \cdots \leq 0 \leq \|e_y - p\|^2.$$

The spectrum of $\mathcal{I}^\Delta(z)$, by our assumption, is

$$\lambda_1 \leq \cdots \leq \lambda_{C-1} \leq \lambda_C.$$

By Hoffman–Wielandt inequality, we have $\forall z \in \Delta^{C-1}, y \in \{1, \cdots, C\}$:

$$\|R(y) - \mathcal{I}^\Delta(z)\| \geq \sqrt{\sum_{i=1}^{C-1} \lambda_i^2 + (\lambda_C - \|e_y - p\|^2)^2}$$

$$\geq |\lambda_C - \|e_y - p\|^2|$$
$$= |\lambda_C - e_y^\top e_y - p^\top p + 2e_y^\top p|$$
$$= |\lambda_C - 1 - \|p\|^2 + 2p_y|$$
$$= \max\{\lambda_C - 1 - \|p\|^2 + 2p_y, \; 1 + \|p\|^2 - \lambda_C - 2p_y\}.$$

By Theorem 1, we have $\lambda_C \leq 1 - \|p\|^2$. One can choose $y$ so that $p_y = p_{(1)}$, then

$$\|R(y) - \mathcal{I}^\Delta(z)\| \geq 1 + \|p\|^2 - \lambda_C - 2p_{(1)}$$
$$\geq 1 + \|p\|^2 - (1 - \|p\|^2) - 2p_{(1)}$$
$$= 2\|p\|^2 - 2p_{(1)}.$$

$\square$

## H    PROOF OF PROPOSITION 5

*Proof.* Similar to Lemma 2, we have

$$\sum_{i=C-k+1}^{C} \lambda_i v_i v_i^\top \preceq \mathcal{I}^\Delta(z) \preceq \text{diag}(p).$$

Therefore

$$\forall x, \theta \quad \sum_{i=C-k+1}^{C} \left(\frac{\partial z}{\partial \theta}\right)^\top \lambda_i v_i v_i^\top \frac{\partial z}{\partial \theta} \preceq \left(\frac{\partial z}{\partial \theta}\right)^\top \mathcal{I}^\Delta(z(x,\theta)) \frac{\partial z}{\partial \theta} \preceq \left(\frac{\partial z}{\partial \theta}\right)^\top \text{diag}(p) \frac{\partial z}{\partial \theta}.$$

Therefore

$$\forall \theta \quad \sum_{x \in \mathcal{D}_x} \sum_{i=C-k+1}^{C} \lambda_i \left(\frac{\partial z}{\partial \theta}\right)^\top v_i v_i^\top \frac{\partial z}{\partial \theta} \preceq \sum_{x \in \mathcal{D}_x} \left(\frac{\partial z}{\partial \theta}\right)^\top \mathcal{I}^\Delta(z(x,\theta)) \frac{\partial z}{\partial \theta} \preceq \sum_{x \in \mathcal{D}_x} \sum_{i=1}^{C} p_i \frac{\partial z_i}{\partial \theta} \frac{\partial z_i}{\partial \theta^\top}.$$

$\square$

## I    PROOF OF COROLLARY 6

*Proof.* We first prove the upper bound. By Proposition 5, we have

$$\mathcal{F}^\Delta(\theta) \preceq \sum_{x \in \mathcal{D}_x} \sum_{i=1}^{C} p_i \frac{\partial z_i}{\partial \theta} \frac{\partial z_i}{\partial \theta^\top}.$$

Taking the trace on both sides, we get

$$\text{tr}(\mathcal{F}^\Delta(\theta)) \leq \sum_{x \in \mathcal{D}_x} \sum_{i=1}^{C} p_i \text{tr}\left(\frac{\partial z_i}{\partial \theta} \frac{\partial z_i}{\partial \theta^\top}\right)$$

$$= \sum_{x \in \mathcal{D}_x} \sum_{i=1}^{C} p_i \text{tr}\left(\frac{\partial z_i}{\partial \theta^\top} \frac{\partial z_i}{\partial \theta}\right)$$

$$= \sum_{x \in \mathcal{D}_x} \sum_{i=1}^{C} p_i \left\|\frac{\partial z_i}{\partial \theta}\right\|^2.$$

The lower bound is not straightforward from Proposition 5. By Eq. (2), we have

$$\text{tr}(\mathcal{F}^\Delta(\theta)) = \sum_{x \in \mathcal{D}_x} \text{tr}\left[\left(\frac{\partial z}{\partial \theta}\right)^\top \mathcal{I}^\Delta(z) \frac{\partial z}{\partial \theta}\right] = \sum_{x \in \mathcal{D}_x} \text{tr}\left[\frac{\partial z}{\partial \theta}\left(\frac{\partial z}{\partial \theta}\right)^\top \mathcal{I}^\Delta(z)\right].$$

Note that $\frac{\partial z}{\partial \theta}\left(\frac{\partial z}{\partial \theta}\right)^\top$ is a $C \times C$ matrix with sorted eigenvalues $\sigma_1^2(x,\theta) \leq \cdots \leq \sigma_C^2(x,\theta)$. By Theorem 1, $\mathcal{I}^\Delta(z)$ is another $C \times C$ matrix with sorted eigenvalues $0 = \lambda_1(x,\theta) \leq \cdots \leq \lambda_C(x,\theta)$. Applying the von Neumann trace inequality, we get

$$\text{tr}(\mathcal{F}^\Delta(\theta)) \geq \sum_{x \in \mathcal{D}_x} \sum_{i=2}^{C} \lambda_i(x,\theta)\sigma_{C-i+1}^2(x,\theta) \geq \sum_{x \in \mathcal{D}_x} \lambda_C(x,\theta)\sigma_1^2(x,\theta).$$

The last "$\geq$" holds because all terms $\lambda_i(x,\theta)\sigma_{C-i+1}^2(x,\theta)$ are non-negative. $\square$

## J    PROOF OF PROPOSITION 7

*Proof.* Denote the singular values of $\frac{\partial z}{\partial \theta}$ as $0 \leq \sigma_1 \leq \cdots \leq \sigma_C$. Then the eigenvalues of the $C \times C$ Hermitian matrix $\frac{\partial z}{\partial \theta}\left(\frac{\partial z}{\partial \theta}\right)^\top$ are $\sigma_1^2 \leq \cdots \leq \sigma_C^2$.

To prove the upper bound, we have

$$\left\| \sum_{x \in \mathcal{D}_x} \sum_{i=1}^{C} p_i \left( \frac{\partial z_i}{\partial \theta} \right)^\top \frac{\partial z_i}{\partial \theta} - \mathcal{F}^\Delta(\theta) \right\|$$

$$= \left\| \sum_{x \in \mathcal{D}_x} \left( \frac{\partial z}{\partial \theta} \right)^\top \left( \operatorname{diag}(p) - \operatorname{diag}(p) + pp^\top \right) \frac{\partial z}{\partial \theta} \right\|$$

$$= \left\| \sum_{x \in \mathcal{D}_x} \left( \frac{\partial z}{\partial \theta} \right)^\top pp^\top \frac{\partial z}{\partial \theta} \right\|$$

$$\leq \sum_{x \in \mathcal{D}_x} \sqrt{ \operatorname{tr}\left[ \left( \frac{\partial z}{\partial \theta} \right)^\top pp^\top \frac{\partial z}{\partial \theta} \left( \frac{\partial z}{\partial \theta} \right)^\top pp^\top \frac{\partial z}{\partial \theta} \right] }$$

$$= \sum_{x \in \mathcal{D}_x} \sqrt{ \operatorname{tr}\left[ p^\top \frac{\partial z}{\partial \theta} \left( \frac{\partial z}{\partial \theta} \right)^\top pp^\top \frac{\partial z}{\partial \theta} \left( \frac{\partial z}{\partial \theta} \right)^\top p \right] }$$

$$\leq \sum_{x \in \mathcal{D}_x} \sqrt{ \left[ p^\top \frac{\partial z}{\partial \theta} \left( \frac{\partial z}{\partial \theta} \right)^\top p \right]^2 }$$

$$= \sum_{x \in \mathcal{D}_x} p^\top \frac{\partial z}{\partial \theta} \left( \frac{\partial z}{\partial \theta} \right)^\top p$$

$$= \sum_{x \in \mathcal{D}_x} \|p\|^2 \cdot \frac{p^\top}{\|p\|} \frac{\partial z}{\partial \theta} \left( \frac{\partial z}{\partial \theta} \right)^\top \frac{p}{\|p\|}$$

$$\leq \sum_{x \in \mathcal{D}_x} \|p\|^2 \sigma_C^2.$$

Now we are ready to prove the lower bound. From the above, we have

$$\left\| \sum_{x \in \mathcal{D}_x} \sum_{i=1}^{C} p_i \left( \frac{\partial z_i}{\partial \theta} \right)^\top \frac{\partial z_i}{\partial \theta} - \mathcal{F}^\Delta(\theta) \right\| = \left\| \sum_{x \in \mathcal{D}_x} \left( \frac{\partial z}{\partial \theta} \right)^\top pp^\top \frac{\partial z}{\partial \theta} \right\|.$$

Denote $\omega(x) := \left( \frac{\partial z}{\partial \theta} \right)^\top p$. Then

$$\left\| \sum_{x \in \mathcal{D}_x} \sum_{i=1}^{C} p_i \left( \frac{\partial z_i}{\partial \theta} \right)^\top \frac{\partial z_i}{\partial \theta} - \mathcal{F}^\Delta(\theta) \right\| = \left\| \sum_{x \in \mathcal{D}_x} \omega(x)\omega(x)^\top \right\|$$

$$= \sqrt{ \operatorname{tr}\left( \left( \sum_{x \in \mathcal{D}_x} \omega(x)\omega(x)^\top \right)^2 \right) }$$

$$\geq \sqrt{ \sum_{x \in \mathcal{D}_x} (\omega(x)^\top \omega(x))^2 }$$

$$= \sqrt{ \sum_{x \in \mathcal{D}_x} \|\omega(x)\|^4 }.$$

The last "$\geq$" follows from

$$\operatorname{tr}\left( \omega(x)\omega(x)^\top \omega(x')\omega(x')^\top \right) = \operatorname{tr}\left( \omega(x')^\top \omega(x)\omega(x)^\top \omega(x') \right) = (\omega(x')^\top \omega(x))^2 \geq 0.$$

$\square$

## K    PROOF OF PROPOSITION 8

*Proof.* First, we can have a loose bound:

$$
\left\| \sum_{x \in \mathcal{D}_x} \sum_{i=C-k+1}^{C} \lambda_i \left(\frac{\partial z}{\partial \theta}\right)^\top v_i v_i^\top \frac{\partial z}{\partial \theta} - \mathcal{F}^\Delta(\theta) \right\|
$$

$$
= \left\| \sum_{x \in \mathcal{D}_x} \sum_{i=C-k+1}^{C} \lambda_i \left(\frac{\partial z}{\partial \theta}\right)^\top v_i v_i^\top \frac{\partial z}{\partial \theta} - \sum_{x \in \mathcal{D}_x} \left(\frac{\partial z}{\partial \theta}\right)^\top \mathcal{I}^\Delta(z) \frac{\partial z}{\partial \theta} \right\|
$$

$$
= \left\| \sum_{x \in \mathcal{D}_x} \left(\frac{\partial z}{\partial \theta}\right)^\top \left(\sum_{i=1}^{C-k} \lambda_i v_i v_i^\top\right) \frac{\partial z}{\partial \theta} \right\|
$$

$$
\leq \left\| \sum_{x \in \mathcal{D}_x} p_{(C-k)} \left(\frac{\partial z}{\partial \theta}\right)^\top \frac{\partial z}{\partial \theta} \right\| \quad \text{(Due to that } \sum_{i=1}^{C-k} \lambda_i v_i v_i^\top \preceq p_{(C-k)} I)
$$

$$
\leq \sum_{x \in \mathcal{D}_x} p_{(C-k)} \left\| \frac{\partial z}{\partial \theta} \left(\frac{\partial z}{\partial \theta}\right)^\top \right\|.
$$

The eigenvalues of $\left(\frac{\partial z}{\partial \theta} \left(\frac{\partial z}{\partial \theta}\right)^\top\right)^2$ are $\sigma_1^4 \leq \cdots \leq \sigma_C^4$. We have

$$
\left\| \left(\frac{\partial z}{\partial \theta}\right)^\top \left(\sum_{i=1}^{C-k} \lambda_i v_i v_i^\top\right) \frac{\partial z}{\partial \theta} \right\|^2
$$

$$
= \text{tr} \left[ \left(\frac{\partial z}{\partial \theta}\right)^\top \left(\sum_{i=1}^{C-k} \lambda_i v_i v_i^\top\right) \frac{\partial z}{\partial \theta} \left(\frac{\partial z}{\partial \theta}\right)^\top \left(\sum_{i=1}^{C-k} \lambda_i v_i v_i^\top\right) \frac{\partial z}{\partial \theta} \right]
$$

$$
= \text{tr} \left[ \left(\frac{\partial z}{\partial \theta} \left(\frac{\partial z}{\partial \theta}\right)^\top \left(\sum_{i=1}^{C-k} \lambda_i v_i v_i^\top\right)\right)^2 \right]
$$

$$
\leq \text{tr} \left[ \left(\frac{\partial z}{\partial \theta} \left(\frac{\partial z}{\partial \theta}\right)^\top\right)^2 \left(\sum_{i=1}^{C-k} \lambda_i^2 v_i v_i^\top\right) \right] \quad \text{(Due to } \text{tr}(AB)^2 \leq \text{tr}(A^2 B^2))
$$

$$
= \text{tr} \left[ \left(\frac{\partial z}{\partial \theta} \left(\frac{\partial z}{\partial \theta}\right)^\top\right)^2 \left(\sum_{i=2}^{C-k} \lambda_i^2 v_i v_i^\top\right) \right] \quad \text{(Note } \lambda_1 = 0)
$$

$$
\leq \sum_{i=2}^{C-k} \sigma_{i+k}^4 \lambda_i^2.
$$

The last "$\leq$" is due to von Neumann's trace inequality. We also have the Cauchy interlacing

$$
\lambda_2 \leq p_{(2)} \leq \lambda_3 \leq p_{(3)} \leq \cdots \leq \lambda_{C-1} \leq p_{(C-1)}.
$$

To sum up,

$$\left\| \sum_{x\in\mathcal{D}_x} \sum_{i=C-k+1}^{C} \lambda_i \left(\frac{\partial z}{\partial \theta}\right)^\top v_i v_i^\top \frac{\partial z}{\partial \theta} - \mathcal{F}^\Delta(\theta) \right\|$$

$$\leq \sum_{x\in\mathcal{D}_x} \left\| \left(\frac{\partial z}{\partial \theta}\right)^\top \left(\sum_{i=1}^{C-k} \lambda_i v_i v_i^\top\right) \frac{\partial z}{\partial \theta} \right\|$$

$$\leq \sum_{x\in\mathcal{D}_x} \sqrt{\sum_{i=2}^{C-k} \sigma_{i+k}^4 \lambda_i^2}$$

$$\leq \sum_{x\in\mathcal{D}_x} \sqrt{\sum_{i=2}^{C-k} \sigma_{i+k}^4 p_{(i)}^2}.$$

If one relaxes $\forall i \in \{2, \cdots, C-k\}, p_{(i)} \leq p_{(C-k)}$, then we get the loose bound proved earlier.

$\square$

## L    PROOF OF PROPOSITION 9

*Proof.*

$$\|\mathcal{F}(\theta) - \overline{\mathcal{F}}^\Delta(\theta)\|_\sigma = \left\| \sum_{x\in\mathcal{D}_x} \left(\frac{\partial z}{\partial \theta}\right)^\top \cdot \mathcal{I}(z(x,\theta)) \cdot \frac{\partial z}{\partial \theta} - \sum_{x\in\mathcal{D}_x} \left(\frac{\partial z}{\partial \theta}\right)^\top (e_y - p)(e_y - p)^\top \frac{\partial z}{\partial \theta} \right\|_\sigma$$

$$= \left\| \sum_{x\in\mathcal{D}_x} \left(\frac{\partial z}{\partial \theta}\right)^\top \left[\mathrm{diag}\,(p) - pp^\top - (e_y - p)(e_y - p)^\top\right] \frac{\partial z}{\partial \theta} \right\|_\sigma$$

$$\leq \sum_{x\in\mathcal{D}_x} \left\| \left(\frac{\partial z}{\partial \theta}\right)^\top \left[\mathrm{diag}\,(p) - pp^\top - (e_y - p)(e_y - p)^\top\right] \frac{\partial z}{\partial \theta} \right\|_\sigma$$

$$\leq \sum_{x\in\mathcal{D}_x} \left\| \frac{\partial z}{\partial \theta} \right\|_\sigma \left\| \mathrm{diag}\,(p) - pp^\top - (e_y - p)(e_y - p)^\top \right\|_\sigma \left\| \frac{\partial z}{\partial \theta} \right\|_\sigma$$

$$= \sum_{x\in\mathcal{D}_x} \sigma_C^2 \left\| \mathrm{diag}\,(p) - pp^\top - (e_y - p)(e_y - p)^\top \right\|_\sigma.$$

Now we examine the matrix $\mathrm{diag}\,(p) - pp^\top - (e_y - p)(e_y - p)^\top$. By Theorem 1, the spectrum of $\mathrm{diag}\,(p) - pp^\top$ is

$$\lambda_1 = 0 \leq \lambda_2 \leq \cdots \leq \lambda_C.$$

By the Cauchy interlacing theorem, the spectrum of $\mathrm{diag}\,(p) - pp^\top - (e_y - p)(e_y - p)^\top$, given by $\lambda_1', \cdots, \lambda_C'$, must satisfy

$$\lambda_1' \leq \lambda_1 = 0 \leq \lambda_2' \leq \lambda_2 \leq \cdots \leq \lambda_C' \leq \lambda_C$$

with at least one non-positive eigenvalue: $\lambda_1' \leq 0$. Therefore

$$\left\| \mathrm{diag}\,(p) - pp^\top - (e_y - p)(e_y - p)^\top \right\|_\sigma \leq \max\{-\lambda_1', \lambda_C\}.$$

We also have

$$\lambda_1' = \inf_{u:\|u\|=1} u^\top \left[\mathrm{diag}\,(p) - pp^\top - (e_y - p)(e_y - p)^\top\right] u$$

$$\geq \inf_{u:\|u\|=1} -u^\top \left[(e_y - p)(e_y - p)^\top\right] u$$

$$= -(e_y - p)^\top (e_y - p)$$

$$= -(1 + p^\top p - 2p_y)$$

$$= 2p_y - 1 - \|p\|^2.$$

Therefore

$$\left\| \operatorname{diag}(p) - pp^\top - (e_y - p)(e_y - p)^\top \right\|_\sigma \leq \max\{1 + \|p\|^2 - 2p_y, \lambda_C\}$$
$$\leq \max\{1 + \|p\|^2 - 2p_y, 1 - \|p\|^2\}$$
$$\leq 1 + \|p\|^2.$$

In summary,

$$\|\mathcal{F}(\theta) - \overline{\mathcal{F}}^\Delta(\theta)\|_\sigma \leq \sum_{x \in \mathcal{D}_x} \sigma_C^2 (1 + \|p\|^2).$$

$\square$

## M   Proof of Proposition 10

*Proof.*

$$\left\| \left(\frac{\partial z}{\partial \theta}\right)^\top \cdot \mathcal{I}^\Delta(z(x, \theta)) \cdot \frac{\partial z}{\partial \theta} - \left(\frac{\partial z}{\partial \theta}\right)^\top \cdot \overline{\mathcal{I}}^\Delta(z(x, \theta)) \cdot \frac{\partial z}{\partial \theta} \right\|_\sigma$$

$$\geq \left\| \left(\frac{\partial z}{\partial \theta}\right)^\top \cdot \left[\mathcal{I}^\Delta(z(x, \theta)) - \overline{\mathcal{I}}^\Delta(z(x, \theta))\right] \cdot \frac{\partial z}{\partial \theta} \right\|_\sigma$$

$$= \left\| \left(\frac{\partial z}{\partial \theta}\right)^\top \cdot \left[\operatorname{diag}(p) - pp^\top - (e_y - p)(e_y - p)^\top\right] \cdot \frac{\partial z}{\partial \theta} \right\|_\sigma$$

$$= \sup_{u:\|u\|=1} \left| \left(\frac{\partial z}{\partial \theta} u\right)^\top \cdot \left[\operatorname{diag}(p) - pp^\top - (e_y - p)(e_y - p)^\top\right] \cdot \left(\frac{\partial z}{\partial \theta} u\right) \right|$$

$$\geq \sup_{v:\|v\|=1} \left| \sigma_{(1)} v \cdot \left[\operatorname{diag}(p) - pp^\top - (e_y - p)(e_y - p)^\top\right] \cdot \sigma_{(1)} v \right|$$

$$\geq \sigma_{(1)}^2 \| \operatorname{diag}(p) - pp^\top - (e_y - p)(e_y - p)^\top \|_\sigma$$

$$\geq \sigma_{(1)}^2 \left| \left(\frac{e_y - p}{\|e_y - p\|}\right)^\top \left((e_y - p)(e_y - p)^\top - \lambda_C\right) \frac{e_y - p}{\|e_y - p\|} \right|$$

$$= \sigma_{(1)}^2 \left| \|e_y - p\|^2 - \lambda_C \right|$$

$$= \sigma_{(1)}^2 \left| 1 + \|p\|^2 - \lambda_C - 2p_y \right|.$$

We choose $p_y = p_{(1)}$, therefore $\exists y$, such that

$$\left\| \left(\frac{\partial z}{\partial \theta}\right)^\top \cdot \mathcal{I}^\Delta(z(x, \theta)) \cdot \frac{\partial z}{\partial \theta} - \left(\frac{\partial z}{\partial \theta}\right)^\top \cdot \overline{\mathcal{I}}^\Delta(z(x, \theta)) \cdot \frac{\partial z}{\partial \theta} \right\|_\sigma$$

$$\geq \sigma_{(1)}^2 \left| 1 + \|p\|^2 - \lambda_C - 2p_{(1)} \right|.$$

$\square$

## N   Proof of Theorem 11

*Proof.* From the derivations in the main text, we already know that $\mathbb{E}_{p(\xi)} \mathbb{F}(\theta) = \mathcal{F}(\theta)$. To show the estimator's variance, we first consider the case when $p(\xi)$ is a standard multivariate Gaussian distribution. Note that both $\mathfrak{h}(\mathcal{D}_x, \theta)$ and $\partial \mathfrak{h}/\partial \theta_i$ are in the form of a sum of independent Gaussian random variables. Hence,

$$\frac{\partial \mathfrak{h}}{\partial \theta_i} = \sum_{x \in \mathcal{D}_x} \sum_{y=1}^{C} \sqrt{p(y \mid x, \theta)} \frac{\partial \ell_{xy}}{\partial \theta_i} \xi_{xy} \sim G\left(0, \sum_{x \in \mathcal{D}_x} \sum_{y=1}^{C} p(y \mid x, \theta) \left(\frac{\partial \ell_{xy}}{\partial \theta_i}\right)^2\right).$$

Therefore

$$\underset{p(\xi)}{\mathbb{E}} \left( \frac{\partial \mathfrak{h}}{\partial \theta_i} \right)^2 = \sum_{x \in \mathcal{D}_x} \sum_{y=1}^{C} p(y \,|\, x, \theta) \left( \frac{\partial \ell_{xy}}{\partial \theta_i} \right)^2 = \mathcal{F}_{ii}(\theta);$$

$$\underset{p(\xi)}{\mathbb{E}} \left( \frac{\partial \mathfrak{h}}{\partial \theta_i} \right)^4 = 3\mathcal{F}_{ii}^2(\theta).$$

Therefore

$$\text{Var}(\mathbb{F}(\theta_i)) = \underset{p(\xi)}{\mathbb{E}} \left( \frac{\partial \mathfrak{h}}{\partial \theta_i} \right)^4 - \mathcal{F}_{ii}^2(\theta) = 2\mathcal{F}_{ii}^2(\theta).$$

We now consider that $p(\xi)$ is Rademacher.

$$\begin{aligned}
\text{Var}(\mathbb{F}(\theta_i)) &= \underset{p(\xi)}{\mathbb{E}} \left( \frac{\partial \mathfrak{h}}{\partial \theta_i} \right)^4 - \left( \mathbb{E} \left( \frac{\partial \mathfrak{h}}{\partial \theta_i} \right)^2 \right)^2 \\
&= \underset{p(\xi)}{\mathbb{E}} \left( \frac{\partial \mathfrak{h}}{\partial \theta_i} \right)^4 - \mathcal{F}_{ii}^2(\theta) \\
&= \underset{p(\xi)}{\mathbb{E}} \left( \sum_{x \in \mathcal{D}_x} \sum_{y=1}^{C} \sqrt{p(y \,|\, x, \theta)} \frac{\partial \ell_{xy}}{\partial \theta_i} \xi_{xy} \right)^4 - \mathcal{F}_{ii}^2(\theta) \\
&= \sum_{x \in \mathcal{D}_x} \sum_{y=1}^{C} p^2(y \,|\, x, \theta) \left( \frac{\partial \ell_{xy}}{\partial \theta_i} \right)^4 \\
&\quad + 3 \sum_{(x,y) \neq (x',y')} p(y \,|\, x, \theta) \left( \frac{\partial \ell_{xy}}{\partial \theta_i} \right)^2 p(y' \,|\, x', \theta) \left( \frac{\partial \ell_{x'y'}}{\partial \theta_i} \right)^2 - \mathcal{F}_{ii}^2(\theta).
\end{aligned}$$

Note that

$$\begin{aligned}
\mathcal{F}_{ii}^2(\theta) &= \left( \sum_{x \in \mathcal{D}_x} \sum_{y=1}^{C} p(y \,|\, x, \theta) \left( \frac{\partial \ell_{xy}}{\partial \theta_i} \right)^2 \right)^2 \\
&= \sum_{x \in \mathcal{D}_x} \sum_{y=1}^{C} p^2(y \,|\, x, \theta) \left( \frac{\partial \ell_{xy}}{\partial \theta_i} \right)^4 + \sum_{(x,y) \neq (x',y')} p(y \,|\, x, \theta) \left( \frac{\partial \ell_{xy}}{\partial \theta_i} \right)^2 p(y' \,|\, x', \theta) \left( \frac{\partial \ell_{x'y'}}{\partial \theta_i} \right)^2.
\end{aligned}$$

Hence,

$$\begin{aligned}
\text{Var}(\mathbb{F}(\theta_i)) &= 3\mathcal{F}_{ii}^2(\theta) - 2 \sum_{x \in \mathcal{D}_x} \sum_{y=1}^{C} p^2(y \,|\, x, \theta) \left( \frac{\partial \ell_{xy}}{\partial \theta_i} \right)^4 - \mathcal{F}_{ii}^2(\theta) \\
&= 2\mathcal{F}_{ii}^2(\theta) - 2 \sum_{x \in \mathcal{D}_x} \sum_{y=1}^{C} p^2(y \,|\, x, \theta) \left( \frac{\partial \ell_{xy}}{\partial \theta_i} \right)^4.
\end{aligned}$$

$\square$

## O  ACCURACY OF HUTCHINSON'S ESTIMATE ON DIAGONAL AND LOW-RANK CORES

In this section, we show that Hutchinson's estimates $\mathbb{F}^{\text{DG}}(\theta)$ and $\mathbb{F}^{\text{LR}}(\theta)$ are both unbiased with bounded variance.

**Proposition 13.** *The random matrix $\mathbb{F}^{\text{DG}}(\theta)$ is an unbiased estimator of $\mathcal{F}^{\text{DG}}(\theta)$. The variance of its diagonal elements is* $\text{Var}\left( \mathbb{F}_{ii}^{\text{DG}}(\theta) \right) = 2(\mathcal{F}_{ii}^{\text{DG}}(\theta))^2 - 2 \sum_{x \in \mathcal{D}_x} \sum_{y=1}^{C} \zeta_y^2(x, \theta)(\frac{\partial z_y}{\partial \theta_i})^4.$

*Proof.*

$$
\underset{p(\xi)}{\mathbb{E}}\left(\mathbb{F}^{\mathrm{DG}}(\theta)\right) = \underset{p(\xi)}{\mathbb{E}}\left(\frac{\partial \mathfrak{h}^{\mathrm{DG}}}{\partial \theta}\frac{\partial \mathfrak{h}^{\mathrm{DG}}}{\partial \theta^{\top}}\right)
$$

$$
= \underset{p(\xi)}{\mathbb{E}}\left(\sum_{x\in\mathcal{D}_x}\sum_{y=1}^{C}\sqrt{\zeta_y(x,\theta)}\frac{\partial z_y}{\partial\theta}\xi_{xy}\sum_{x'\in\mathcal{D}_x}\sum_{y'=1}^{C}\sqrt{\zeta_{y'}(x',\theta)}\frac{\partial z_{y'}}{\partial\theta^{\top}}\xi_{x'y'}\right)
$$

$$
= \sum_{x\in\mathcal{D}_x}\sum_{y=1}^{C}\sum_{x'\in\mathcal{D}_x}\sum_{y'=1}^{C}\sqrt{\zeta_y(x,\theta)}\sqrt{\zeta_{y'}(x',\theta)}\frac{\partial z_y}{\partial\theta}\frac{\partial z_{y'}}{\partial\theta^{\top}}\underset{p(\xi)}{\mathbb{E}}\left(\xi_{xy}\xi_{x'y'}\right)
$$

$$
= \sum_{x\in\mathcal{D}_x}\sum_{y=1}^{C}\zeta_y(x,\theta)\frac{\partial z_y}{\partial\theta}\frac{\partial z_y}{\partial\theta^{\top}}
$$

$$
= \sum_{x\in\mathcal{D}_x}\left(\frac{\partial z}{\partial\theta}\right)^{\top}\mathcal{I}^{\mathrm{DG}}(z(x,\theta))\frac{\partial z}{\partial\theta}
$$

$$
= \mathcal{F}^{\mathrm{DG}}(\theta).
$$

Therefore,

$$
\underset{p(\xi)}{\mathbb{E}}\left(\mathbb{F}_{ii}^{\mathrm{DG}}(\theta)\right) = \underset{p(\xi)}{\mathbb{E}}\left(\frac{\partial \mathfrak{h}^{\mathrm{DG}}}{\partial\theta_i}\right)^{2} = \sum_{x\in\mathcal{D}_x}\sum_{y=1}^{C}\zeta_y(x,\theta)\left(\frac{\partial z_y}{\partial\theta_i}\right)^{2} = \mathcal{F}_{ii}^{\mathrm{DG}}(\theta).
$$

$$
\underset{p(\xi)}{\mathbb{E}}\left(\frac{\partial \mathfrak{h}^{\mathrm{DG}}}{\partial\theta_i}\right)^{4} = \underset{p(\xi)}{\mathbb{E}}\left(\sum_{x\in\mathcal{D}_x}\sum_{y=1}^{C}\sqrt{\zeta_y(x,\theta)}\frac{\partial z_y}{\partial\theta_i}\xi_{xy}\right)^{4}
$$

$$
= \sum_{x\in\mathcal{D}_x}\sum_{y=1}^{C}\zeta_y^{2}(x,\theta)\left(\frac{\partial z_y}{\partial\theta_i}\right)^{4} + 3\sum_{(x,y)\neq(x',y')}\zeta_y(x,\theta)\left(\frac{\partial z_y}{\partial\theta_i}\right)^{2}\zeta_{y'}(x',\theta)\left(\frac{\partial z_{y'}}{\partial\theta_i}\right)^{2}
$$

$$
= 3(\mathcal{F}_{ii}^{\mathrm{DG}}(\theta))^{2} - 2\sum_{x\in\mathcal{D}_x}\sum_{y=1}^{C}\zeta_y^{2}(x,\theta)\left(\frac{\partial z_y}{\partial\theta_i}\right)^{4}.
$$

Hence,

$$
\mathrm{Var}(\mathbb{F}_{ii}^{\mathrm{DG}}(\theta)) = \underset{p(\xi)}{\mathbb{E}}\left(\frac{\partial \mathfrak{h}^{\mathrm{DG}}}{\partial\theta_i}\right)^{4} - (\mathcal{F}_{ii}^{\mathrm{DG}}(\theta))^{2}
$$

$$
= 2(\mathcal{F}_{ii}^{\mathrm{DG}}(\theta))^{2} - 2\sum_{x\in\mathcal{D}_x}\sum_{y=1}^{C}\zeta_y^{2}(x,\theta)\left(\frac{\partial z_y}{\partial\theta_i}\right)^{4}.
$$

$\square$

**Proposition 14.** $\mathbb{F}^{\mathrm{LR}}(\theta)$ *is an unbiased estimate of* $\mathcal{F}^{\mathrm{LR}}(\theta)$; *the variance of its diagonal elements is*
$\mathrm{Var}\left(\mathbb{F}_{ii}^{\mathrm{LR}}(\theta)\right) = 2(\mathcal{F}_{ii}^{\mathrm{LR}}(\theta))^{2} - 2\sum_{x\in\mathcal{D}_x}\lambda_C^{2}(x,\theta)\left(v_C^{\top}(x,\theta)\frac{\partial z}{\partial\theta_i}\right)^{4}.$

*Proof.* The proof is similar to Proposition 13 and is also based on Hutchinson's trick.

$$\mathop{\mathbb{E}}_{p(\xi)} \left( \mathbb{F}^{\mathrm{LR}}(\theta) \right)$$

$$= \mathop{\mathbb{E}}_{p(\xi)} \left( \frac{\partial \mathfrak{h}^{\mathrm{LR}}}{\partial \theta} \frac{\partial \mathfrak{h}^{\mathrm{LR}}}{\partial \theta^{\top}} \right)$$

$$= \mathop{\mathbb{E}}_{p(\xi)} \left( \sum_{x \in \mathcal{D}_x} \sqrt{\lambda_C(x,\theta)} \left( \frac{\partial z}{\partial \theta} \right)^{\top} v_C(x,\theta) \xi_x \sum_{x' \in \mathcal{D}_x} \sqrt{\lambda_C(x',\theta)} v_C(x',\theta)^{\top} \left( \frac{\partial z}{\partial \theta} \right) \xi_{x'} \right)$$

$$= \sum_{x \in \mathcal{D}_x} \lambda_C(x,\theta) \left( \frac{\partial z}{\partial \theta} \right)^{\top} v_C(x,\theta) v_C(x,\theta)^{\top} \left( \frac{\partial z}{\partial \theta} \right)$$

$$= \mathcal{F}^{\mathrm{LR}}(\theta).$$

Therefore

$$\mathop{\mathbb{E}}_{p(\xi)} \left( \mathbb{F}^{\mathrm{LR}}_{ii}(\theta) \right) = \sum_{x \in \mathcal{D}_x} \lambda_C(x,\theta) \left( \left( \frac{\partial z}{\partial \theta_i} \right)^{\top} v_C(x,\theta) \right)^2 = \mathcal{F}^{\mathrm{LR}}_{ii}(\theta);$$

$$\mathop{\mathbb{E}}_{p(\xi)} \left( \frac{\partial \mathfrak{h}^{\mathrm{LR}}}{\partial \theta_i} \right)^4 = \mathop{\mathbb{E}}_{p(\xi)} \left( \sum_{x \in \mathcal{D}_x} \sqrt{\lambda_C(x,\theta)} v_C^{\top}(x,\theta) \frac{\partial z}{\partial \theta_i} \xi_x \right)^4$$

$$= \sum_{x \in \mathcal{D}_x} \lambda_C^2(x,\theta) \left( v_C^{\top}(x,\theta) \frac{\partial z}{\partial \theta_i} \right)^4$$

$$+ 3 \sum_{x \neq x'} \lambda_C(x,\theta) \left( v_C^{\top}(x,\theta) \frac{\partial z}{\partial \theta_i} \right)^2 \lambda_C(x',\theta) \left( v_C^{\top}(x',\theta) \frac{\partial z}{\partial \theta_i} \right)^2$$

$$= 3 (\mathcal{F}^{\mathrm{LR}}_{ii}(\theta))^2 - 2 \sum_{x \in \mathcal{D}_x} \lambda_C^2(x,\theta) \left( v_C^{\top}(x,\theta) \frac{\partial z}{\partial \theta_i} \right)^4.$$

Hence,

$$\mathrm{Var} \left( \mathbb{F}^{\mathrm{LR}}_{ii}(\theta) \right) = \mathop{\mathbb{E}}_{p(\xi)} \left( \frac{\partial \mathfrak{h}^{\mathrm{LR}}}{\partial \theta_i} \right)^4 - (\mathcal{F}^{\mathrm{LR}}_{ii}(\theta))^2$$

$$= 2 (\mathcal{F}^{\mathrm{LR}}_{ii}(\theta))^2 - 2 \sum_{x \in \mathcal{D}_x} \lambda_C^2(x,\theta) \left( v_C^{\top}(x,\theta) \frac{\partial z}{\partial \theta_i} \right)^4.$$

$$\square$$

We have $\mathrm{Std}(\mathbb{F}^{\mathrm{DG}}_{ii}(\theta))/\mathcal{F}^{\mathrm{DG}}_{ii}(\theta) \leq \sqrt{2}$ by Proposition 13, and at the same time, we have $\mathrm{Std}(\mathbb{F}^{\mathrm{LR}}_{ii}(\theta))/\mathcal{F}^{\mathrm{LR}}_{ii}(\theta) \leq \sqrt{2}$ by Proposition 14.

Thus both estimators have theoretical guarantees on their estimation quality (CV bounded by $\sqrt{2}$).

