# OpenReview forum: "Deterministic Bounds and Random Estimates of Metric Tensors on Neuromanifolds"
_ICLR.cc/2026/Conference — ICLR 2026 Poster_

### Official Review · Reviewer_bqwM · 2025-10-29

**Soundness:** 4
**Presentation:** 3
**Contribution:** 3
**Rating:** 6
**Confidence:** 2

**Summary:**

This work presents a new approach along with theoretical results for estimating the Fisher Information Matrix (FIM) in deep neural networks. The deterministic upper and lower bounds were examined via the geometric properties in low-dimensional probability simplex spaces, followed by extending to high-dimensional parameter spaces using pullback metrics. This estimator can be computed with just a single backward pass, providing significant computational benefits over existing Monte Carlo and empirical FIM methods for large networks like DistilBERT.

**Strengths:**

1. Theoretical results are sufficient and rigorous.

2. The ease of implementation in standard deep learning frameworks, which is evident by the empirical validation with better MAE while keeping computational costs similar. The framework's independence from architecture and its ability to apply to both diagonal and low-rank approximations offer flexible trade-offs between accuracy and computational cost.

**Weaknesses:**

1. Empirical validation relies solely on DistilBERT and the results from two text classification datasets. How computational overhead scales with network depth and width, memory needs for large models?

**Questions:**

None

---

> ### Author Response · Authors · 2025-11-20
>
> Thank you for your thoughtful review and recognizing the strengths. We have updated the manuscript accordingly.
>
> #### **Limited empirical validation**
>
> We have extended our test-bed to six dataset-model combinations
> including NLP, vision, and audio tasks, with a strengthened evaluation protocol.
> For example, we have included RoBERTa-base (>0.1B parameters) on Multi-Genre Natural Language
> Inference (MNLI). Our baselines now range from a small number of classes (SST-2, C=2)
> to a moderate number of classes (CIFAR-100, C=100).
> AG News (C=4) has been replaced by a more difficult NLP dataset, DBpedia (C=14).
> In the latest experiments, we further confirm the initial observation
> that *Hutchinson's estimate (with one probe) is as fast as empirical FIM and is more accurate.*
> We kindly refer the reviewer to the updated experiments section on pages 8--9.
>
> #### **Computational overhead w.r.t. network depth and width**
>
> The main computational cost of the Hutchinson estimate
> involves a standard backward pass through the computational graph
> (which is already built during the forward pass).
> Therefore, its **computational complexity matches that of
> a single backward pass** ($O(\dim(\theta))$),
> identical to the empirical FIM.
> Our low-rank variants ($\mathbb{F}^{LR(k)}$) require eigendecompositions
> of $C \times C$ matrices and therefore are more expensive.
> In the experiments, they run at roughly half the speed of the empirical FIM.
>
> #### **Memory needs for large models**
>
> To compute the diagonal FIM using Hutchinson's estimate $\mathbb{F}$
> requires a float32 buffer to store all the diagonal FIM elements.
> Therefore, the memory overhead scales with $O(\dim(\theta))$,
> the total number of parameters. Again, this is the same as the empirical FIM.

---

> > ### Comment · Reviewer_bqwM · 2025-11-27
> >
> > Thank you for the response. I have no further concerns.

---

### Official Review · Reviewer_rVQk · 2025-11-01

**Soundness:** 3
**Presentation:** 3
**Contribution:** 2
**Rating:** 4
**Confidence:** 3

**Summary:**

This paper proposes deterministic bounds and Hutchinson-based stochastic estimators for the Fisher Information Matrix (FIM) on neural manifolds. The method provides unbiased, bounded-variance FIM estimates requiring only one backward pass, making it scalable to large networks. Experiments on DistilBERT show feasible and stable approximations compared with empirical and Monte Carlo estimators.

**Strengths:**

- Provides a unified and theoretically grounded framework for deterministic and stochastic FIM estimation with clear variance guarantees.
- The Hutchinson estimator is efficient and easy to integrate into deep learning pipelines, enabling scalable information-geometric analysis.

**Weaknesses:**

- The paper does not validate the estimator on analytically known probability distributions, especially in high-dimensional settings. This makes it difficult to verify the claimed accuracy of the proposed bounds and stochastic estimates.
- The practical impact of improved FIM accuracy on downstream tasks (e.g., optimization, generalization) is not demonstrated. In some real applications [a], approximation may even be preferable to precision, since neural networks themselves are inherently approximate. The authors should strengthen the discussion and empirical evidence on how their estimator benefits real learning tasks.

[a] Deep CNNs Meet Global Covariance Pooling: Better Representation and Generalization

**Questions:**

see wk

---

> ### Author Response · Authors · 2025-11-20
>
> Thank you for your constructive review and for recognizing the strengths. We have prepared a revision based on your comments.
>
> #### **Validate the estimators on their accuracy**
>
> We have expanded and refined the experimental section, with NLP, vision, and audio tasks. In particular,
> in our latest experiments we compute the **ground truth** using automatic differentiation.
> Under the current protocol,
> obtaining this ground truth is **computationally intensive** and infeasible in practice.
> It requires between $10^4$ and $10^6$ backward passes, depending on the task.
> We report the *relative MAE*, which is a more meaningful error metric
> because it normalizes the error w.r.t. the ground truth.
> We find that Hutchinson's unbiased estimate consistently achieves the
> lowest error across all dataset-model combinations,
> whereas the widely adopted empirical FIM exhibits a substantial bias.
>
> #### **Practical impact of improved FIM accuracy**
>
> This paper focuses on the fundamental problem of estimating the FIM, which
> underlies applications in neural network optimization, pruning,
> model identifiability, and information-geometric analysis.
> This focus enables a deeper and more general theoretical treatment,
> which leads to our core contributions.
> In response to the reviewer's suggestion, we
> have updated the introduction with the following clarifications
> on **practical implications**:
>
> *Given its deep and broad background, estimating $\mathcal{F}(\theta)$
> with guaranteed quality is important
> even in the absence of a specific application pipeline.
> Inaccurate estimates can lead to overly aggressive or overly conservative
> learning steps, or miscalculated saliency scores
> and suboptimal pruning decisions.
> In learning theory, a loosely estimated FIM
> undermines the validity of geodesic distances
> and the applicability of Cram\'er-Rao lower bounds,
> and may distort curvature-based sharpness,
> which is closely linked to generalization.*

---

> ### Comment · Reviewer_rVQk · 2025-11-21
>
> Thanks for the responses and additional results.
>
> Although this is not my core area of expertise, the response convince me that such estimators could be quite useful. I have raise my rating from 4 to 6 accordingly.
>
> One minor suggestion for the rebuttal phase: it may be beneficial to highlight the modified parts in the revised PDF (e.g., using colored text). This would make it easier for other reviewers to quickly locate the changes and follow the updates more efficiently.
>
> Good luck

---

> > ### Author Response · Authors · 2025-11-21
> >
> > Thank you for recognizing our revision and the potential usefulness of the proposed technique.
> >
> > We apologize for the not highlight the modified parts -- the paper is updated accordingly, with the major changes highlighted in pink (for only the main text in the first 10 pages).
> >
> > Furthermore, we have added a new section B in the appendix to explain an alternative implementation of Hutchinson's FIM estimation.

---

### Official Review · Reviewer_u7Fy · 2025-11-07

**Soundness:** 3
**Presentation:** 3
**Contribution:** 2
**Rating:** 6
**Confidence:** 3

**Summary:**

The authors propose an unbiased estimator for the Fisher metric in the parameter space of a neural network, in the setting of multiclass classification, where the underlying geometry is determined by the simplex. The main estimator, along with two additional approximations, is straightforward to compute and leverages automatic differentiation. The authors provide a theoretical analysis to analyze the accuracy and some properties of the proposed estimators. The theoretical claims are further supported by empirical demonstrations.

**Strengths:**

- The problem of approximating the Fisher metric in neural network settings is timely and very important. The proposed estimators are simple and can be easily computed in practice.
- The technical part of the paper appears to be sound and the theoretical claims correct, but I have not followed every proof in detail. The experiments support the theoretical analysis.
- The paper is generally well written and accessible.

**Weaknesses:**

- Since the paper proposes an estimator for the Fisher metric, I would expect additional empirical demonstrations regarding the estimator’s accuracy and efficiency. I acknowledge that the theoretical results constitute the primary contribution, but I believe that further benchmark experiments would help demonstrate the properties of the estimator and also consider comparisons to related methods. The current experimental section provides some insight, but additional comparisons would strengthen the paper.
- As the authors mention in Section 5, Hutchinson-type estimators have been used in related contexts. While the extension to neural networks is appreciated and well motivated, as a non-expert in Fisher information analysis, I cannot argue which of the theoretical contributions are novel and which may already be known.

**Questions:**

Q1. It would be helpful to include a comparison with a Monte Carlo approach in terms of both computational complexity and accuracy.

Q2. Is it correct that the LB estimator typically performs better when the predicted distribution approaches one of the corners of the simplex (see Lemma 3)?

Q3. I suggest including the true Fisher information in the experimental comparison, at least in settings where it is feasible.

Q4. A simple synthetic example where the full Fisher information matrix can be computed and visualized would help demonstrate the behavior of the estimator.

---

> ### Author Response · Authors · 2025-11-20
>
> Thank you for your insightful and constructive review. We have updated our submission accordingly. In below, we first address your explicit questions and then the raised weaknesses.
>
> #### **Q1. Compare with Monte Carlo (MC) approach**
>
> The MC estimator requires one backward pass *per sample*,
> whereas our Hutchinson estimate requires one backward pass *per batch*.
> These backward passes are the computational bottleneck of both methods.
> Thus, the Hutchinson estimate
> achieves a speed up proportional to the batch size.
> This also explains why MC is rarely used in production settings.
>
> In terms of accuracy, Hutchinson's estimate has bounded variance
> (Proposition 11),
> while the naive MC estimator lacks comparable guarantees.
> Across all tasks in our experiments,
> the vanilla Hutchinson estimate can approximate the true
> diagonal FIM with a typical relative error of $\pm 20\\%$ (measured by relative mean absolute error, RelMAE).
>
> #### **Q2. LB estimator performs better (as in Lemma 3)**
>
> Yes, the LB (low-rank, denoted as $\mathbb{F}^{LR}$)
> estimator performs better when the output distribution is close to one-hot.
> As shown on SST-2 and MNLI in Table 2, the fine-tuned models
> produce low-entropy outputs, under which the LB estimator excels.
> This is also shown in Proposition 8.
> However, LB estimators require computing the eigenvectors of
> the $C\times C$ core FIM, making them slightly more expensive
> than the vanilla Hutchinson estimate for a small to moderate $C$ and infeasible for a large $C$.
>
> #### **Q3. Include the true Fisher information**
>
> We now include the true FIM across all tasks.
> Note this is very expensive to compute and is only feasible on a subset
> (we use 128 batches, with a batch size of 64).
> In the extreme case (ResNet-50/EfficientNet-B0 on CIFAR-100),
> computing the exact FIM on this subset requires around $8 \times 10^5$ backward passes.
>
> #### **Q4. A simple synthetic example**
>
> Our experiments prioritize realistic production-oriented settings,
> whose FIM spectra differs substantially from toy examples.
> For each dataset-model combination, we provide
> (1) accuracy w.r.t. the true FIM; (2) speed w.r.t. the empircial FIM; (3) visualization of the FIM distribution,
> decomposed across successive neural network components.
>
> #### **W1. Empirical accuracy/efficiency; Further benchmarks**
>
> We have extended our test-bed to include 6 dataset-model pairs,
> with NLP, vision, and audio tasks, and with a strengthened evaluation protocol.
> We evaluate both the RelMAE and the speed up factor w.r.t. empirical FIM.
> In the latest experiments, we further confirm the initial observation
> that *Hutchinson's estimate (with one probe) is as fast as empirical FIM and is more accurate.*
> For more details, we kindly refer the reviewer to the updated experiments section on pages 8--9.
>
> #### **W2. Novelty of Hutchinson-type estimators and theoretical contributions**
>
> Indeed, the Hutchinson's trick has been used in related contexts as discussed
> in the last paragraph of the Related Work section.
> Our estimator **differs from prior efforts in three essential aspects**
> (1) it is unbiased to the true FIM;
> (2) it comes with explicit variance bounds;
> (3) it is scalable to modern architectures.
> Crucially, no previous method simultaneously satisfies all three properties.
>
> Our theoretical analysis is novel in that our bound quality characterizations
> rely on matrix perturbation theory, and our results are formulated in terms of
> order statistics of the network output, so one can argue how the sparsity of the
> output probability vector improves the bound quality.

---

### Official Review · Reviewer_YTgp · 2025-11-10

**Soundness:** 3
**Presentation:** 4
**Contribution:** 3
**Rating:** 8
**Confidence:** 3

**Summary:**

The paper analyzes the Fisher Information Matrix (FIM) for neural classifiers through deterministic bounds based on spectral analysis and proposes a novel Hutchinson-based estimator. The theoretical development is rigorous with complete proofs in the appendix, though experimental validation remains limited.

**Strengths:**

Solid mathematical foundation: All major theorems have complete proofs in the appendix. The spectral analysis of the simplex FIM (Theorem 1) and envelope characterizations (Lemma 2) are rigorously established.

Novel computational approach: The Hutchinson estimator with “detach-and-mix” construction provides unbiased FIM estimates with provably bounded diagonal CV ≤ √2, addressing variance issues in Monte Carlo methods.

Comprehensive theoretical analysis: The paper provides both deterministic bounds and stochastic estimates with detailed variance analysis for Gaussian and Rademacher distributions.

**Weaknesses:**

Experiments: Only DistilBERT tested on AG News and SST-2. Missing: (i) comparison with K-FAC, diagonal empirical Fisher (Adam), exact Gauss-Newton; (ii) optimization performance metrics; (iii) other architectures (CNNs, ResNets). Just I guess a general expansion in this area would be nice, though you note each formulation may require a from-scratch derivation for each. Please comment on this

Unclear practical advantage: No demonstration of how the estimator improves optimization or learning dynamics compared to existing methods.

Terminology issue: “Singular semi-Riemannian metric” is incorrect I believe - the FIM is degenerate PSD, not indefinite.

**Questions:**

Technical Concerns

Computational complexity underspecified: Power iteration convergence for computing λ_C, v_C could be slow near uniform distributions (small spectral gap). The O(MC|𝒟_x|) cost needs quantification.
Implementation gaps: No guidance on choosing probe count for target accuracy, mini-batch handling, or numerical stability considerations.

Minor Issues

Notation overload (multiple uses of F̂ for both empirical and MC Fisher)
Some bounds are loose (e.g., diagonal upper bound always has error ≥ 1/C). Correct me if im wrong/misinterpret

Questions for Authors

What is the wall-clock time comparison with K-FAC for equivalent accuracy?
Can you demonstrate improved optimization performance in a practical setting?
How does mini-batching affect the variance bounds?

Verdict

The paper makes solid theoretical contributions with rigorous mathematical development. However, it lacks the experimental validation needed to demonstrate practical value. The theory is great and extensive, but the work feels a little missing in terms of comprehensive experiments showing real optimization improvements. But I am also open to understanding if this mainly geared to a paper of more theoretical build up value (or step in that direction) than a fully practical ready usage.

---

> ### Author Response · Authors · 2025-11-21
>
> Thank you for your comprehensive and insightful review. We have prepared a revision accordingly.
>
> #### **Comparison with empirical Fisher, exact Gauss-Newton**
>
> We have included diagonal Empirical Fisher, and included
> the exact diagonal Fisher in the updated experimental section.
> Due to the high cost of the exact diagonal Fisher (bottleneck of our experimental pipeline),
> we use a random subset (fixed for all FIM estimators) of 128 batches, each of size 64 for all datasets.
> The exact Fisher allows us to quantitatively measure the accuracy of different estimators.
>
> #### **Missing other architectures**
>
> In the updated manuscript, besides DistilBERT, we have included RoBERTa,
> ResNet-50, EfficientNet-B0, Wav2Vec-2, covering NLP/Vision/Audio tasks.
>
> #### **Comprehensive experiments**
>
> We have enhanced our experimental section to include
>
> - Accuracy comparison (w.r.t. the ground truth)
>
> - Runtime comparison
>
> - More salable architectures and more data samples used to compute the FIM
>
> #### **Optimization improvements and compare with K-FAC**
>
> This paper focuses on the **core problem of estimating the FIM**
> without committing to a specific application pipeline.
> The FIM underlies applications in neural network optimization, pruning,
> model identifiability, and information-geometric analysis.
> This focus enables a deeper and more general theoretical treatment,
> which leads to our core contributions.
>
> We acknowledge that the natural gradient and its variants are among the
> most significant applications of the FIM to neural networks.
> Incorporating our estimator into an optimizer requires additional engineering beyond the scope of the paper:
> (1) accumulating and damping mechanisms;
> (2) Efficient inversion of the FIM (e.g. based on Sherman–Morrison updates).
> From our view, these aspects would constitute a distinct line of work.
>
> Hutchinson's estimate is unbiased, whereas K-FAC is biased (approximation).
> Our current diagonal FIM estimator is scalable to large-scale architectures,
> while the vanilla K-FAC is known to become costly for transformer-based networks.
> As such, our work serves as a fundamental building block of future applications,
> which can be combined with other techniques (e.g. K-FAC).
>
> Following the reivewer's suggestion, we have further clarified that estimating
> $\mathcal{F}$ with guaranteed quality is the core topic, and how an inaccurate
> estimate could affect applications (see the 3rd paragraph of the introduction).
>
> #### **Notation of empirical and MC Fisher**
>
> The updated paper now uses different notations.
>
> #### **Some bounds are loose**
>
> We can confirm the reviewer's observation: the diagonal upper bound is
> relatively loose, although it still beats empirical FIM in most of the test
> cases (see table 2).  We have improved our low-rank lower bound to use $k$
> dominant eigen-vectors of the core (rather than 1 in the original submission).
> It is much tighter in theory (see updated Proposition 8) and in the experiments.
>
> #### **Power iteration convergence slow**
>
> The reviewer is correct on that the convergence can be slow when the FIM core matrix has a small spectral gap.
> Our results on random classification head (DBpedia, CIFAR-100(ResNet-50), CIFAR-100(EfficientNet-B0), SpeechCommands in Table 2)
> further confirm that $\mathbb{F}^{LR}$ is not preferred for high-entropy/small-eigengap outputs.
> We have revised the last paragraph of section 4.4, to clarify that the convergence depends on the spectral gap.
>
> We also include an estimator $\mathbb{F}^{\mathrm{LR(k)}}$ which uses PyTorch's eigh() interface rather than power iterations to contract with $\mathbb{F}^{\mathrm{LR}}$.
>
> #### **Mini-batching affect the variance bounds**
>
> Our variance bound reduces as the batch size decreases.
> We have added a remark of Proposition 11 (page 7).
> At the dataset level, the variance is inversely proportional to $J$,
> while the computational grows linearly with $J$, presenting a typical
> accuracy-computation trade-off.
>
> #### **Singular semi-Riemannian metric**
>
> The reviewer is correct that the FIM is degenerate in general.
> Our terminology follows Kupeli's *Singular Semi-Riemannian Geometry*.
> In that framework, a singular semi-Riemannian metric is a broad class,
> which when non-degenerate becomes a semi-Riemannian metric (e.g. Minkowski),
> and this in turn includes Riemannian metrics.

---

### Meta-Review · Area_Chair_cazW · 2026-01-08

**Summary:**

The paper analyzes the Fisher Information Matrix (FIM) for neural networks for classification through deterministic bounds based on spectral analysis and proposes a novel Hutchinson-based estimator. Extensive theoretical analysis is carried out, accompanied by numerical experiments.

Reviewers generally appreciate the solid and comprehensive theoretical analysis carried out in the paper and the authors carried out further numerical experiments to support the theoretical claims. Overall, the results should be of sufficient interest to the community.

Note: I agree with Reviewer YTgp that the terminology "singular semi-Riemannian metric" is not correct. The FIM is generally a positive semi-definite matrix. It is not indefinite.

**Reviewer Concerns:**

Reviewers are generally positive about the paper

- One main concern was the limited numerical experiments (Reviewers YTgp, u7Fy,  bqwM, rVQk).
In their rebuttal, the authors added additional experiments for accuracy comparison (w.r.t. the ground truth), runtime comparison, more architectures and  data samples used to compute the FIM.

**Reviewer Scores:**

The scores are 8,6,4,6

It is likely that Reviewers would have increase their scores to vote for Accept.

---

### Decision · Program_Chairs · 2026-01-26

Accept (Poster)